

# Conditions favorable for secondary ice production in Arctic mixed-phase clouds

Julie T. Pasquier[1], Jan Henneberger[1], Fabiola Ramelli[1], Annika Lauber[1,*], Robert O. David[2], Jörg Wieder[1], Tim Carlsen[2], Rosa Gierens[3], Marion Maturilli[4], and Ulrike Lohmann[1]

[1]Institute for Atmospheric and Climate Science, ETH Zürich, Zurich, Switzerland
[2]Department of Geosciences, University of Oslo, Oslo, Norway
[3]Institute for Geophysics and Meteorology, University of Cologne, Cologne, Germany
[4]Alfred Wegener Institute, Helmholtz Centre for Polar and Marine Research (AWI), Potsdam, Germany
[*]Now at: Center for Climate Systems Modelling (C2SM), ETH Zürich, Zurich, Switzerland

**Correspondence:** Julie T. Pasquier (julie.pasquier@env.ethz.ch) and Jan Hennberger (jan.henneberger@env.ethz.ch)

**Abstract.** The Arctic is very susceptible to climate change and thus warming much faster than the rest of the world. Clouds influence terrestrial and solar radiative fluxes, and thereby impact the amplified Arctic warming. The partitioning of thermo-dynamic phases (i.e. ice crystals and water droplets) within mixed-phase clouds (MPCs) especially influences their radiative properties. However, the processes responsible for ice crystal formation remain only partially characterized. In particular, so-called secondary ice production (SIP) processes, which create supplementary ice crystals from primary ice crystals and the environmental conditions that they occur in, are poorly understood. The microphysical properties of Arctic MPCs were measured during the Ny-Ålesund AeroSol Cloud ExperimENT (NASCENT) campaign to obtain a better understanding of the atmospheric conditions favorable for the occurrence of SIP processes. To this aim, the in-situ cloud microphysical properties retrieved by a holographic cloud imager mounted on a tethered balloon system were complemented by ground-based remote sensing and ice nucleating particle measurements. During six days investigated in this study, SIP occurred during 40% of the in-cloud measurements and high SIP events with number concentrations larger than 10 L$^{-1}$ of small pristine ice crystals in 3.5% of the in-cloud measurements. This demonstrates the role of SIP for Arctic MPCs. The highest concentrations of small pristine ice crystals were produced at temperatures between -3 °C and -5 °C and were related to the occurrence of drizzle drops freezing upon collision with ice crystals. This suggests that a large fraction of ice crystals in Arctic MPCs is produced via the droplet shattering mechanism. From evaluating the ice crystal images, we could identify ice-ice collision as a second SIP mechanism that dominated when fragile ice crystals were observed. Moreover, SIP occurred over a large temperature range and was observed in up to 95% of the measurements down to -24 °C due to the occurrence of ice-ice collisions. This emphasizes the importance of SIP at temperatures below -8 °C, which are currently not accounted for in most numerical weather models.

## 1 Introduction

Clouds influence the radiation budget in two competing ways. On the one hand, they scatter shortwave radiation back to space and thereby cool the surface. On the other hand, they absorb and re-emit longwave radiation and thereby warm the surface.



The Arctic is experiencing an amplified warming (Meredith et al., 2019), which is influenced by several feedback processes associated with temperature, water vapour, and clouds (Goosse et al., 2018). The influence of clouds on the radiation budget in the Arctic is especially complex and uncertain in because of the strongly varying reflection from the surface below (sea

ice or water) or the lack of solar radiation during polar night (e.g., Goosse et al., 2018). In addition, the phase partitioning and concentration of the cloud particles determine the exact radiative properties of the mixed-phase clouds (MPCs) consisting of water vapor, cloud droplets, and ice crystals (Sun and Shine, 1994). Because the cloud particle concentration and phase partitioning strongly influence the radiative properties of MPCs, a thorough understanding of the processes that determine the formation and transformation of cloud particles is required.

At temperatures below -38 °C, cloud droplets freeze homogeneously, whereas at temperatures between -38 °C and 0 °C, primary ice crystals form on ice nucleating particles (INPs). However, many observations have shown that the ice crystal number concentration (ICNC) in MPCs is frequently several orders of magnitude higher than the measured INP concentration (INPC) (e.g., Hobbs and Rangno, 1985, 1998; Ladino et al., 2017; Korolev et al., 2020). This discrepancy can be explained by additional ice crystals falling from a seeder cloud aloft (Proske et al., 2021), by the influence of surface processes such as

blowing snow (e.g. Beck et al., 2018), or by the formation of secondary ice crystals from the existing ice crystals (e.g., Hallett and Mossop, 1974; Takahashi et al., 1995; Field et al., 2017; Korolev and Leisner, 2020). This last process, known as secondary ice production (SIP), is thought to play a critical role in the formation of ice crystals in supercooled clouds (e.g., Korolev et al., 2020; Korolev and Leisner, 2020).

Several SIP mechanisms have been proposed over the past decades: droplet shattering during freezing, rime-splintering

during riming, fragmentation during ice-ice collision, and fragmentation during sublimation (e.g. Field et al., 2017; Korolev and Leisner, 2020). Droplet shattering is defined as the ejection of secondary ice crystals caused by cracking, fragmentation, bubble bursting or jetting, which can occur due to pressure build-up during freezing of droplets (e.g., Mason and Maybank, 1960; Takahashi and Yamashita, 1970; Lauber et al., 2018; Keinert et al., 2020). The rime-splintering or Hallett-Mossop process (Hallett and Mossop, 1974; Mossop, 1978) refers to the production of secondary ice during riming and is expected to

occur when cloud droplets freeze upon collision with large rimed ice particles (e.g., Mossop, 1978, 1985; Field et al., 2017; Korolev and Leisner, 2020). Fragmentation during collision of several ice particles can lead to their fragmentation, which creates secondary ice crystals (Vardiman, 1978; Takahashi et al., 1995). Finally, fragmentation during ice crystal sublimation in unsaturated regions can create secondary ice crystals, but it requires the re-entering of the fragments back into saturated cloud regions, otherwise the complete sublimation of the fragment is likely (Dong et al., 1994; Bacon et al., 1998).

The environmental conditions favorable for SIP were mostly assessed in laboratory studies (see Korolev and Leisner, 2020, for an overview of laboratory studies on SIP). Temperature, cloud droplet concentrations and sizes, and ice crystal sizes and habits are particularly relevant for the occurrence of SIP (e.g., Korolev and Leisner, 2020). The temperature range between -3 °C and -8 °C was suggested to be the most favorable for the occurrence of rime-splintering (Hallett and Mossop, 1974; Mossop and Hallett, 1974), whereas the maximum rate of fragments produced by droplet shattering or by ice-ice collision

was observed at around -15 °C in laboratory studies (Takahashi and Yamashita, 1970; Takahashi et al., 1995; Lauber et al., 2018). However, evidence for droplet shattering has been observed over a much wider temperature range, from -20 °C up to



-0.5 °C during field observations of natural MPCs (Korolev et al., 2020; Lauber et al., 2021; Pasquier et al., 2022a) and in laboratory experiments (Keinert et al., 2020). Cloud droplets are needed for the rime-splintering and the droplet shattering processes. Although droplets smaller than 12 μm and larger than 24 μm are necessary for the rime-splintering process (e.g.,

Mossop, 1978, 1985; Korolev and Leisner, 2020), the probability for droplet shattering occurrence increases with increasing droplet size (Lauber et al., 2018; Keinert et al., 2020). The size and concentration of the droplets is in turn influenced by aerosol particles acting as cloud condensation nuclei (CCN), by updrafts, by the general cloud dynamics, and by the cloud lifetime (Lohmann et al., 2016). The ice crystal number concentrations and the ice crystal shapes and sizes are also relevant for SIP. In particular, large rimed ice crystals were found to increase the rate of splinters ejected during rime-splintering (Hallett and

Mossop, 1974) and ice-ice collision (Vardiman, 1978).

However, there are large inconsistencies and many gaps in current knowledge of the physical mechanisms and environmental conditions favourable for SIP due to the scarcity of laboratory and field measurements (Korolev and Leisner, 2020). In addition, direct measurements of SIP processes in-cloud are challenging as the secondary fragments and splinters of a few micrometers or less are typically below the resolution limit of cloud measurement probes and the probability of observing a cloud particle

when it is involved in SIP is infinitesimally small. Furthermore, the presence of an INP in ice particles can only be determined on a crystal by crystal basis, which requires that each ice crystal is sampled and analyzed individually for the presence of an INP (Hoffer and Braham, 1962; Mertes et al., 2007; Worringen et al., 2015; Mignani et al., 2019). However, when the concentration of small ice crystals exceeds that of ambient INPs, SIP processes must have contributed to the ICNC. As such, several studies compare INPC with total ICNC to infer the occurrence of SIP (e.g., Ladino et al., 2017; Li et al., 2021; Wieder

et al., 2022a). The cloud microphysical properties can additionally be used to identify the mechanism potentially responsible for SIP. For example, rimed particles together with a sufficient concentration of cloud droplets (with diameter below 12 μm and above 24 μm) at temperatures between -8 °C and -3 °C are an indicator for the occurrence of the rime-splintering process (e.g., Lloyd et al., 2015). Meanwhile, drizzle drops and/or frozen drops can be indicators for the occurrence of droplet shattering (e.g., Lawson et al., 2017), and large rimed particles or broken ice crystals at relatively low temperatures may be indicators

for ice-ice collisions.

Even if SIP parametrisations were used on case studies for the ice-ice collision and droplet shattering mechanisms (e.g., Sotiropoulou et al., 2020; Dedekind et al., 2021; Georgakaki et al., 2022), only the rime-splintering process is widely used in numerical weather and climate models. However, an accurate description of SIP processes and of the environmental conditions favorable for SIP is needed to correctly represent the phase partitioning within MPCs to estimate their radiative properties in

the Arctic (Young et al., 2019).

The present study aims to identify conditions favorable for SIP in low-level Arctic MPCs using a holographic imager mounted on the tethered balloon system HoloBalloon (Ramelli et al., 2020), together with ground-based INP and remote sensing measurements. The results presented originate from six days of measurement in MPCs collected during the Ny-Ålesund AeroSol Cloud ExperimENT (NASCENT) campaign (Pasquier et al., 2022a) in Ny-Ålesund, Svalbard. First, the main instru-

mentation and the methodology applied for SIP identification are described in Section 2. Second, we present the meteorology and the occurrence of SIP during six measurement days in Section 3. Then, the environmental conditions associated with the





SIP occurrence are then examined in Section 4. Lastly, the final remarks and recommendations for future work are given in Section 5.

## 2 Methods

### 2.1 Measurement location

The data presented in this paper was collected during the NASCENT campaign, which took place in Ny-Ålesund, Svalbard, (78.9° N, 11.9° E, Fig. 1a) from September 2019 to August 2020 with the goal to enhance the existing knowledge about aerosols and clouds in the Arctic climate, and their interactions throughout the year. A description of the campaign and the main instrumentation is given in Pasquier et al. (2022a). Ny-Ålesund is situated on the south side of Kongsfjorden and on the northern side of a mountain range, with Mt. Zeppelin as the closest mountain 2.5 km southeastward of the settlement (Fig. 1b). The surface wind is strongly influenced by the topography (Fig. 1b) and is typically channelled along Kongsfjorden (Beine et al., 2001; Maturilli et al., 2013; Maturilli and Kayser, 2017; Pasquier et al., 2022a).

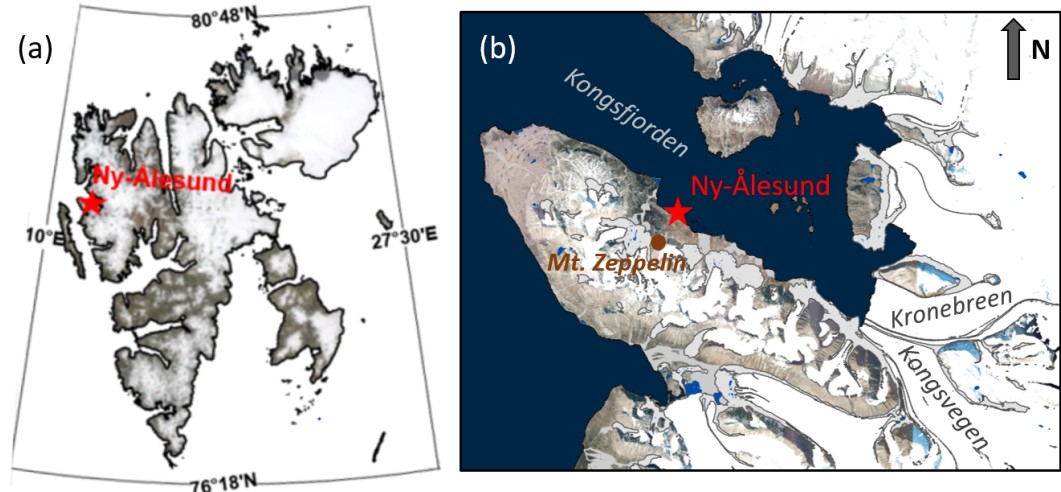

**Figure 1.** (a) Map of Svalbard with the location of Ny-Ålesund marked with the red star. (b) Map of the peninsula close to Ny-Ålesund. Ny-Ålesund, the Kronebreen and Kongsvegen glaciers, the fjord Kongsfjorden, and the Mt. Zeppelin mountain are labelled. (Topographical data from  Norwegian Polar Institute, 2014).

### 2.2 Instrument setup

The tethered balloon system HoloBalloon (Ramelli et al., 2020) was used to perform in-situ cloud microphysical measurements during October - November 2019 and March - April 2020. HoloBalloon consists of a cloud measurement platform hanging 12 m below a helikite. The main instrument on the measuring platform is the HOLographic cloud Imager for Microscopic





Objects (HOLIMO). HOLIMO images cloud particles in the size range from small cloud droplets (6 μm) to precipitation-sized particles (2 mm) in a three-dimensional sample volume to obtain information about the phase-resolved particle size distribution and particle habits (Henneberger et al., 2013; Beck et al., 2017; Ramelli et al., 2020). The classification of cloud

droplets and ice crystals is performed based on their shape using a convolutional neural network trained and fine-tuned on cloud particles from holographic imagers (Touloupas et al., 2020; Lauber, 2020). The smallest detectable ice crystals are 25 μm and all particles below this threshold are automatically classified as cloud droplets. Furthermore, ice crystals with a rather circular shape in the 2D image are misclassified as cloud droplets. All ice crystals were manually classified into habits based on their 2D shape to plates, columns, frozen drops, recirculation particles (see Section3.2 for details), and aged particles that comprise

rimed, aggregated, and irregular ice crystals. In addition, cloud droplets and artefacts wrongly classified as ice crystals by the convolutional neural network were manually reclassified. Therefore, the uncertainty in the concentration of ice particles can be estimated with ±5% for ice crystals smaller than 100 μm and ±15% for ice crystals larger than about 100 μm (Beck, 2017). For cloud droplets, the uncertainty is estimated to be ±6% as determined for the classification with the convolutional neural network in Touloupas et al. (2020). The sampling volume of HOLIMO is about 16-20 cm$^3$ per frame, and approximately

4-6 frames were taken per second, which gives a volume of 3 L to 60 L for the averages over 30 s to 5 min used in this study. Thus, the limit of detection of HOLIMO, corresponding to one cloud particle measured in the time average, amounts to ∼0.3 L$^{-1}$ for measurements averaged over 30 seconds. Note that using a tethered balloon system such as HoloBalloon for cloud microphysical measurements has the advantage that the influence from ice crystals lifted from the ground (e.g., blowing snow, Beck et al., 2018) can be neglected due to the distance of the measurements from the surface. In addition, thanks to the

low true air speed of HOLIMO on the tethered balloon system and the adequate tower tips, the shattering of ice crystals in the sample volume is minimized.

Ambient aerosol was sampled through a heated inlet mounted on top of an observatory container located next to the launching location of HoloBalloon (Pasquier et al., 2022a). Downstream the inlet, a high flow-rate impinger (Coriolis® μ, Bertin Instruments, France) operating at 300 L min$^{-1}$ collected ambient aerosol particles with aerodynamic diameter of 0.5 μm and

larger into pure water. For one sample, the impinger collected aerosol particles for one hour, probing a volume of 18 m$^3$. Directly after collection, each sample was analysed for INPC via the offline technique DRoplet Ice Nuclei Counter Zurich (DRINCZ, David et al., 2019), which measured INPC at sub-freezing temperatures between approximately -3 °C and -20 °C. INPCs were calculated according to Vali (1971), corrected for the sampling water's background, converted to concentration in air, and their uncertainties were calculated applying Gaussian error propagation. Further details of the processing are pre-

sented in Wieder et al. (2022b) and Li et al. (2022). The lower INPC detection limit amounts to 1.4·10$^{-4}$ L$^{-1}$ and the relative measurement uncertainty is on average given by a factor of two.

The in-situ measurements were complemented by remote sensing instruments installed at the French–German Arctic Research Base AWIPEV. In particular, the 94 GHz cloud radar of University of Cologne (JOYRAD-94, Küchler et al., 2017) was used for analyzing the whole cloud structure, the ceilometer (Vaisala-CL51, Maturilli and Ebell, 2018) was utilized to

determine the cloud base height, and the wind lidar (Windcube200) enabled the continuous characterisation of wind direction and speed in the lower troposphere. Meteorological surface measurements were continuously available from the AWIPEV





observation site (Maturilli et al., 2013, 2015) and the vertical atmospheric structure was determined by daily and additional radiosondes (Maturilli and Kayser, 2017) during specific measurement periods.

### 2.3 SIP identification

We use a specific method to identify cloud regions where SIP was recently occurring from in-situ measurements, using the concentration of small pristine ice crystals (diameters <100 µm) following the approach introduced by Korolev et al. (2020). This approach is based on the fact that if SIP occurs in a supersaturated environment, the newly formed ice fragments or splinters rapidly grow by water vapor diffusion into detectable faceted ice crystal habits representative of the environment in which they grow in (e.g, Nakaya, 1954; Libbrecht, 2005). With time, the ice crystal habit can lose its spatial correlation with its

environment of origin due to turbulent diffusion, horizontal and/or vertical advection. Korolev et al. (2020) estimated the time for which a secondary ice particle remains associated with its environment of origin to be 60-120 s, which allows a hexagonal plate or column to grow to a width or length between 50 µm and 150 µm at water saturation, depending on its aspect ratio and the environmental temperature. Following this method, we use the occurrence of pristine ice crystals with a major axis between 25 µm and 106 µm as an indicator for SIP regions. The major axis is defined as the major axis of an ellipse that

encompasses the detected pixels of the particle. This specific cut-off size was chosen as it is the bin size of the size distribution used in the processing of the data closest to 100 µm, thus lying in between 50 µm and 150 µm. Examples of pristine ice crystals smaller than 106 µm, used as indicators for SIP regions, are shown in Figure 2 and contrasted with non-pristine smaller than 106 µm and pristine ice crystals larger than 106 µm. Non-pristine crystals cannot have formed from vapor deposition growth, and could originate from breakups during impact with the instrument payload or from rime falling from the tethered balloon.

Such particles were therefore excluded from the SIP analysis.

The identified SIP regions were further classified into three SIP classes, namely, low SIP regions ($SIP_{low}$), moderate SIP regions ($SIP_{mod}$), and high SIP regions ($SIP_{high}$) using the number concentration of pristine ice crystals with diameters < 106 µm ($ICNC_{pr<106\ \mu m}$) as follows:

(1) $SIP_{low}$: $0.3\ L^{-1} \leq ICNC_{pr<106\ \mu m} < 1\ L^{-1}$,

(2) $SIP_{mod}$: $1\ L^{-1} \leq ICNC_{pr<106\ \mu m} \leq 10\ L^{-1}$,

(3) $SIP_{high}$: $ICNC_{pr<106\ \mu m} \geq 10\ L^{-1}$.

In addition, $SIP_{all}$ represents the three SIP classes combined and $SIP_{no}$ refers to $ICNC_{pr<106\ \mu m} < 0.3\ L^{-1}$, with $0.3\ L^{-1}$ being the lower limit of detection of HOLIMO for measurements averaged over 30 s. This means that if no small pristine ice crystals is measured, the actual $ICNC_{pr<106\ \mu m}$ is below $0.3\ L^{-1}$ but not necessarily $0\ L^{-1}$. This signifies that all the $ICNC_{pr<106\ \mu m}$ smaller

than $0.3\ L^{-1}$ are not taken into account in the analysis of SIP in this study. Note that the contribution from primary ice nucleation in the remote Arctic region around Ny-Ålesund is expected to be lower than this $0.3\ L^{-1}$ at temperatures above -20 °C (e.g., Tobo et al., 2020; Rinaldi et al., 2021; Li et al., 2022).





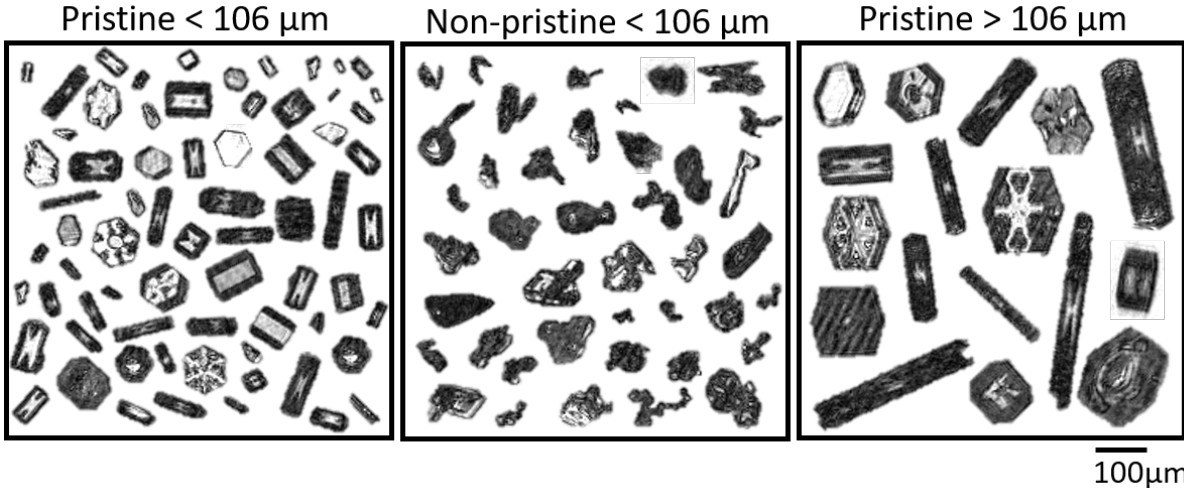

**Figure 2.** Examples of ice crystals observed with HOLIMO classified as pristine with diameters < 106 μm , non-pristine ice crystals with diameters < 106 μm, and pristine ice crystals with diameters > 106 μm. The presence of pristine ice crystals with diameter < 106 μm was used for identification of SIP. The scale bar applies to all panels.

To ensure that the measurements were conducted in-cloud, only regions where the relative humidity with respect to ice derived from the interpolated radiosonde measurements is higher than 95% or the liquid water content measured by HOLIMO was larger than 0.005 g m$^{-3}$ are taken into account. Both criteria are used disjointly because in some cases the cloud may only be saturated with respect to ice, and in other cases the relative humidity measured by the radiosonde closest in time may not be capturing local areas of saturation.

### 2.4 Determination of INP concentrations

The INPC derived from the DRINCZ measurements on the ground is used to estimate the INPC at the cloud top (INPC$_{CT}$) and at the HoloBalloon measurement altitude (INPC$_{HB}$). As the INPC is a function of the nucleation temperature (increasing exponentially with decreasing temperature), we use the temperatures at cloud top and at the measurement location of HoloBalloon to estimate INPC$_{CT}$ and INPC$_{HB}$. These temperatures are derived from the linearly interpolated radiosonde temperature profiles together with the highest cloud top altitude retrieved by the cloud radar on each day and the measurement altitude of HoloBalloon (see Section A2 in the Appendix for details). INPC$_{CT}$ represents the cloud's highest INPC estimate as the lowest cloud temperatures are generally found at cloud top. INPC$_{CT}$ is therefore representative for the maximum ICNC that could have formed via primary nucleation from INPs. INPC$_{HB}$ is representative for the ICNC that could have formed by primary nucleation on INPs at the measurement location and can be directly compared to ICNC$_{pr<106\mu m}$ because the method employed assumes that the ice crystals smaller 106 μm have formed close to HoloBalloon's location.

Uncertainties arise from using INP measurements taken at the surface to estimate the in-cloud INPC. For well-mixed boundary layers, in which the aerosol particle concentrations are constant between the surface and cloud base, the INPC at the ground





and in the cloud should be comparable (neglecting INP depletion by scavenging and INP entrainment at cloud top). However, in decoupled cloud cases, when a shear layer and/or a large potential temperature increase is observed below the cloud base, the INPC in the cloud could be different than the one observed at the ground. In the cases presented in this study, the layers from cloud base to the surface were generally well-mixed and no strong decoupling case was observed (Fig. A1). In addition,

Pasquier et al. (2022a) compared the INPC measured at the observatory container at sea level on 12 November 2019 and the INPC averaged over several days at the mountaintop Zeppelin Observatory located 2 km southwestward at 475 m a.s.l. (Fig. 1b) and found that the INPC were in agreement within a factor of 5 at the two location despite the different measurement method and time averages used.

## 3  SIP occurrence during six days of MPC measurements in Arctic MPCs

### 3.1  Overview of the six days with MPCs

The microphysical properties of the MPCs were identified with HOLIMO on five consecutive days from 8 to 12 November 2019 and on 1 April 2020. The total cloud droplet number concentrations (CDNC) measured by HOLIMO reached up to 30 cm$^{-3}$ and drizzle drops (defined with diameter larger than 64 µm) were observed during four measurement flights (Fig. 3d). This CDNC is considerably lower than for comparable continental clouds, which typically have CDNCs of up to 1000 cm$^{-3}$

(Lohmann et al., 2016), but is representative for the pristine Arctic environment where limited CCN availability results in low CDNCs, as discussed in e.g., Lance et al. (2011) and Koike et al. (2019). Generally, ICNC$_{pr<106 µm}$ is orders of magnitude larger than INPC$_{HB}$ and ICNC is orders of magnitude larger than INPC$_{CT}$, except on 10 November 2019 (Fig. 3e). This indicates that primary ice nucleation via INPs cannot be solely responsible for the observed ICNC, and suggests that SIP processes contributed to the ICNC.

On 8 November 2019, an occluded front moved over Ny-Ålesund, producing strong southwesterly large-scale winds (up to 20 m s$^{-1}$ at 2000 m a.s.l.) and about 12 mm of accumulated precipitation (not shown). As the front passed, the low-level cloud field was overrun by a deep cloud layer that extended to cloud top temperatures below -38 °C at an altitude of 5000 m a.s.l.. At these temperatures, any cloud droplet would freeze independently of INPs via homogeneous freezing. On 9 November 2019, the sea level pressure dropped by about 7 hPa and the surface wind speed increased from 2 to 8 m s$^{-1}$ as another low pressure system

passed over Ny-Ålesund (Fig. 3a,b). During the flights performed on 8 and 9 November 2019, HoloBalloon measured mostly in subsaturated regions below cloud, where the cloud droplets and ice crystals were evaporating and sublimating, respectively, as also indicated by the relative humidity below 100% below ∼700 m observed by the radiosondes (Fig. 4). Evidence of ice crystal sublimation can been deduced from the rounded edges of the ice crystals and the thin filaments connecting parts of the crystals to their main body (Fig. 5a). It is evident that such ice crystals could easily break up in two or more particles

depending on their original shape, thereby creating secondary ice crystals. However, unless these fragments were reintroduced into regions with ice (super)saturation by updrafts, they will sublimate completely.







**Figure 3.** (a) Ambient temperature and pressure measured from the weather mast two meters above ground at the AWIPEV Observatory. (b) Horizontal wind speed measured with the wind lidar averaged over 1h (wind barbs) and HoloBalloon measurement height (black line). (c) Cloud radar reflectivity (color), HoloBalloon measurement height (black line), and cloud top temperatures from radiosonde launches measured during the six-day measurement period. On 8 November 2019 and 1 April 2020 the temperature is shown at an altitude of 1800 m a.s.l. because the cloud top is higher than 3000 m a.s.l.. (d) Total CDNC (black) and drizzle drops number concentration (DDNC) (orange) averaged over 5 min. The uncertainty in the concentration of cloud and drizzle is estimated to be ±6%. (e) Total ICNC (black line) and $\text{ICNC}_{pr<106\,\mu m}$ (red line) averaged over 5 min, $\text{INPC}_{CT}$ (light blue crosses) and $\text{INPC}_{HB}$ (dark blue crosses). For 10 November 2019, the ICNCs averaged over each flight are shown with black circles because the ICNC are too low to display a time series. On 12 November 2019, the $\text{INPC}_{HB}$ were below the limit of detection of the INP instrumentation, therefore the limit of detection ($1.4 \cdot 10^{-4}$ L$^{-1}$) is displayed instead ($\text{INPC}_{lim}$, dark blue dashed line). The uncertainty for the concentration of ice particles smaller than 100 μm is estimated to ±5% and to ±15% for ice crystals larger than 100 μm. The uncertainty for the INPC amounts to a factor of two. On 8 November 2019 and 1 April 2020, no $\text{INPC}_{CT}$ can be provided as the cloud top temperatures were below the observable nucleation temperatures of our INP instrumentation. All data are shown from 11:00 UTC on 8 November to 18:00 UTC on 12 November 2019 and on 1 April 2020 from 05:00 to 16:00 UTC. Note that the ticks are at 12:00 UTC for each day.

On 8 and 9 November 2019, updrafts estimated from the remote sensing observation at the HoloBalloon location (see Appendix A3 for the methods) reached up to 2.5 m s$^{-1}$ and 1 m s$^{-1}$, respectively. These moderate updrafts could have lifted some fragments back into ice supersaturated regions, where they could have grown again and increased the ICNC. Otherwise,
if the ice crystals sublimated completely, the remaining INPs could have re-entered the cloud and formed new ice crystals (e.g., Solomon et al., 2015; Possner et al., 2017; Fu et al., 2019). Although this could act as a pathway to enhance ICNCs, the resulting ice formation mechanism would be primary ice crystal nucleation and not SIP.

After the low pressure system moved eastward of Ny-Ålesund on 10 November 2019, the flow became northwesterly and advected cold air towards Ny-Ålesund. This cold northwesterly flow pushed under the warmer air that was present in the
fjord valley before, and by that acted like a cold front lifting the air and causing the formation of a shallow and very lightly precipitating stratocumulus cloud deck. Consistently, the temperature at the surface dropped from approximately -3 °C to -10 °C within a few hours (Fig. 3a). Two measurement flights were conducted on 10 November 2019 and HoloBalloon was able to penetrate through the cloud deck with cloud top temperature of -17 °C (Figs. 3c and 4). The CDNCs measured by HOLIMO were about 20-30 cm$^{-3}$ (Fig. 3d). A few dendrite-like ice crystals were measured by HOLIMO during both flights (Fig. 5b)
and the ICNC averaged over the entire flight period amounted to $1.4 \cdot 10^{-2}$ L$^{-1}$ (Fig. 3e). No pristine ice crystals smaller than 106 μm were measured and the mean ICNC lies in the daily variability of the $\text{INPC}_{CT}$ observed (Fig. 3e). Thus, we conclude that the ice crystals formed by primary nucleation on INPs and that no SIP process substantially increased the ICNCs on this day. Therefore, the INP availability determined the ice crystal formation. This shows the ability of INPs to control ice crystal formation in remote pristine areas like the Arctic in case of shallow clouds and weak dynamics.





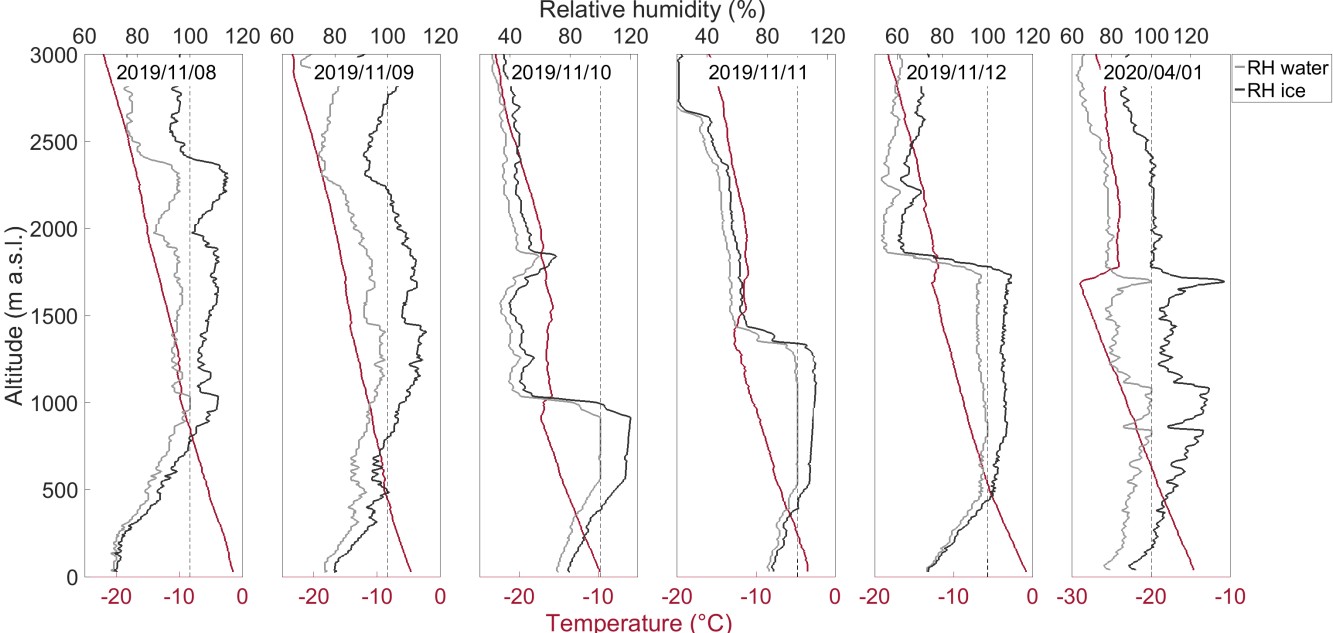

**Figure 4.** Temperature (red) and relative humidity (RH) with respect to water (bright grey) and ice (dark grey) measured by the radiosonde launched at 11:00 UTC on 8 to 12 November 2019 and at 17:00 UTC on 1 April 2020. The 100% RH line is shown with the broken black line.

On 11 and 12 November 2019, the weather in Ny-Ålesund was influenced by the passage of a warm front embedded with several precipitation showers. In these two days, the MPC evolved from a $SIP_{low}$ state with $ICNC_{pr<106\,\mu m}$ below 1 $L^{-1}$ to a $SIP_{high}$ state with $ICNC_{pr<106\,\mu m}$ greater than 50 $L^{-1}$. As this is about 5 orders of magnitude higher than the estimated $INPC_{HB}$, we propose that SIP mechanisms were responsible for the sudden increase in $ICNC_{pr<106\,\mu m}$ and examine the contribution from the likely active SIP processes in detail in Section 3.2.

On 1 April 2020, a warm front passed over Ny-Ålesund and produced a cirrostratus cloud at 8000 m. This cirrostratus deepened to an an altostratus deck that acted as a *seeder* cloud that precipitated into the low-level mixed-phase *feeder* cloud below, thereby enhancing the ICNC in the low-level MPC measured by HoloBalloon. However, the $INPC_{HB}$ was up to 1 to 2 orders of magnitude smaller than the $ICNC_{pr<106\,\mu m}$, which indicates that some SIP processes were likely active in the low-level MPC. The microphysical properties of the low-level mixed-phase feeder cloud are discussed in Section 3.3.





8-9 November 2019          10 November 2019

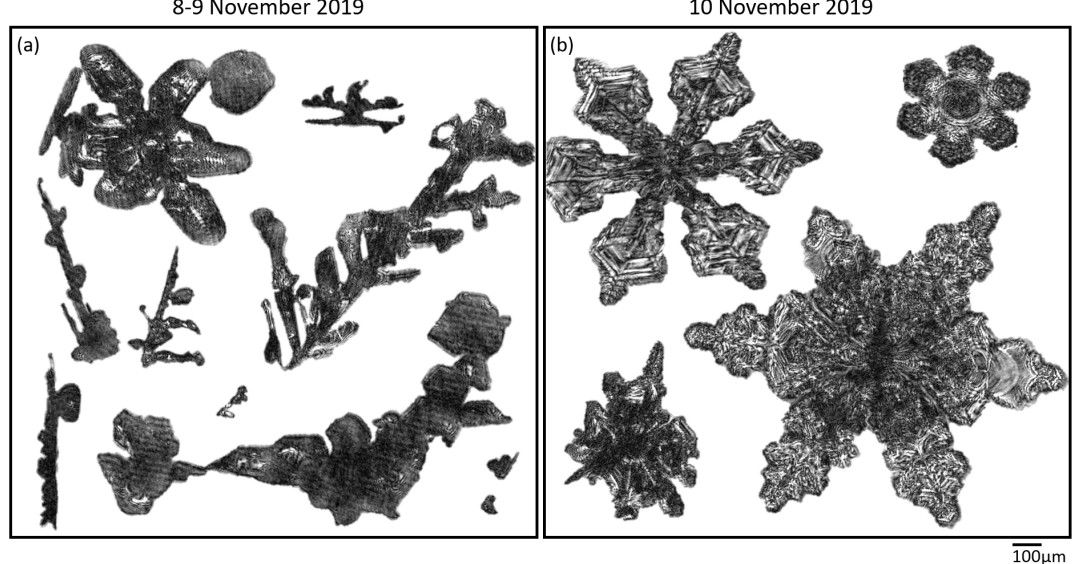

100μm

**Figure 5.** Representative examples of ice crystals observed with HOLIMO during the flights on (a) 8 and 9 November 2019 and (b) 10 November 2019. The scale bar applies to both panels.

## 3.2 High SIP event on 11 November 2019

On 11 November 2019, a precipitating low-level MPC was observed with a cloud base around 700 m a.s.l. and cloud top rising from about 1000 m a.s.l. to 2200 m a.s.l. (Fig. 6a). The surface temperature increased from -3.1°C to -0.3°C between 11:00 UTC and 20:00 UTC (Fig. 3a), whereas the cloud top temperature decreased from -11°C C to -13.5°C as the cloud top height increased. The cloud radar observed regions of enhanced reflectivity, indicative of the presence of large ice crystals (Fig. 6a). Two flights were performed at 10:15-13:40 UTC and 15:50-19:00 UTC into the MPC with HoloBalloon (Fig. 6a). The measured cloud droplet size distribution peaked at around 50 μm and drizzle drops were observed, except for a short period between 13:15 and 13:45 UTC when the CDNC spectra peaked at smaller sizes (Fig. 6b).

The measured cloud evolved from low total ICNC ranging between 0.3 and 10 L$^{-1}$ and ICNC$_{pr<106\,\mu m}$ below 1 L$^{-1}$ during the first flight (10:15-13:40 UTC), to a region with total ICNC ranging mostly between 5 and 20 L$^{-1}$ and ICNC$_{pr<106\,\mu m}$ between 1-3 L$^{-1}$ (contributing about 3-30% to total ICNC) (15:50-18:10 UTC) and finally to a region with ICNC up to 150 L$^{-1}$, out of which up to 90 L$^{-1}$ (60%) were ICNC$_{pr<106\,\mu m}$ (18:10-18:45 UTC) (Figs. 6c and 7b,c). This last period (18:10-18:45 UTC) is marked by several peaks in ICNC above 100 L$^{-1}$ and ICNC$_{pr<106\,\mu m}$ above 10 L$^{-1}$ (Figs. 6c and 7b). On this day, the INPC$_{CT}$ varied between $1 \cdot 10^{-3}$ and $9 \cdot 10^{-3}$ L$^{-1}$ and the INPC$_{HB}$ between $1 \cdot 10^{-4}$ and $4 \cdot 10^{-4}$ L$^{-1}$ (Fig. 3e), thus four to five orders of magnitude lower than the ICNC and ICNC$_{pr<106\,\mu m}$. No increase in INPC is observed during the course of the day. Hence, nucleation on INPs cannot explain the measured peaks in ICNC$_{pr<106\,\mu m}$ at 18:10 UTC onwards. Therefore, we assign the increases to local SIP processes.





**Figure 6.** Overview of the cloud properties on 11 November 2019. (a) Cloud radar reflectivity (color), HoloBalloon measurement height (black line), cloud base height measured by the ceilometer (black dots), and temperatures at the corresponding altitudes measured by the radiosonde at 11:00 UTC, 14:00 UTC, and 20:00 UTC. Note that the lowering of the cloud base to the surface detected by the ceilometer after 17:30 UTC is caused by precipitation. (b) Cloud droplet size distributions (color shading) and total CDNC (black line). The uncertainty in the concentration of cloud and drizzle is estimated to be ±6%. (c) Ice crystal size distributions (color shading) and total ICNC (black line) measured by HOLIMO averaged over 1 min. The uncertainty for the concentration of ice particles smaller than 106 μm is estimated to ±5% and for the concentration of larger ice crystals to ±15%.





Locally formed ice crystals smaller than 106 μm were mostly elongated columns with a large aspect ratio between 3 and 9 (Fig. 7a). These habits are consistent with the environmental temperature (-4.5 °C) at their measurement location. The high aspect ratio of the columns indicates that the cloud layer had a relatively high water supersaturation (Nakaya, 1954; Libbrecht,

2005). Note that columns with a maximum length larger than 106 μm were observed (see Fig. 7a) but not accounted for in the ICNC$_{pr<106\,μm}$.

Ice crystal habits help to understand which SIP processes contributed to the increase in ICNC$_{pr<106\,μm}$. Ice crystals observed during SIP periods were frozen drops, aged particles, and recirculated particles (Fig. 7a), which are a mix of columnar and plate-like crystals due to the crystals growing in different temperature regimes (Korolev et al., 2020; Pasquier et al., 2022b).

The observation of frozen drops during SIP periods suggests that the droplet shattering process produced splinters during the freezing of drizzle drops (e.g., Lauber et al., 2018; Korolev and Leisner, 2020). In particular, the ratios of frozen drops to total ICNC were especially large (0.6) at 18:05-18:10 UTC just before the first and largest peak in ICNC$_{pr<106\,μm}$ (Fig. 7c). Coincidentally, some observed frozen drops were identified on HOLIMO images to have accreted with small columns, suggesting that the collision of drizzle drops with ice crystals initiated their freezing.

A likely explanation for this first ICNC$_{pr<106\,μm}$ peak is therefore that the droplet shattering mechanism caused the formation of splinters which grew to small pristine columns. Then these small columns could collide with further drizzle drops, thereby initiating their freezing and the formation of additional ice splinters. This could have led to a cascading SIP process via a positive feedback loop that can explain the rapid increase in ICNC$_{pr<106\,μm}$, as already proposed by Lawson et al. (2015). The fraction of frozen drops is lower after this peak in ICNC$_{pr<106\,μm}$ at 18:10 UTC (Fig. 7c) and the concentration of large drops

decreased after this peak as well (Fig. 6b), indicating that the drizzle drops froze and precipitated out of the cloud. Thus, we propose that droplet shattering was largely contributing to the peak of ICNC$_{pr<106\,μm}$ (90 L$^{-1}$) at 18:10 UTC.

Between 18:20 and 18:55 UTC, droplet shattering plays likely a lesser role. Instead, SIP by ice-ice collision seem to dominate after recirculation particles appear to concentration up to 10 L$^{-1}$ after 18:15 UTC (Fig. 7b,c). As these particles contained fragile branches, their collision and subsequent break-up could created additional ice crystals. Indeed, particles resembling

broken branches were observed (highlighted with the dark brown box in Figure 7a). The fraction of recirculation particles to ICNC is especially large between 18:20 and 18:45 UTC. Therefore, we suggest that the ice-ice collision break-up contributed to the peaks in ICNC$_{pr<106\,μm}$ observed during this period together with droplet shattering.

The temperature was in the range of the rime-splintering process, however the CDNC was between 0.1 and 3 cm$^{-3}$ between 18:10 and 18:45 UTC (Fig. 6b), and the concentration of cloud droplets smaller than 12 μm required for the rime-splintering

process (Mossop, 1978, 1985) was between 0.01 and 0.2 cm$^{-3}$. Thus, the probability of collision of rimed particles with droplets at these small concentrations is likely too low to have any important effect on the rime-splintering process. Earlier on this day, the aged/rimed particles were the most frequent ice crystal observed (Figure 7c) and the CDNCs (Fig. 6b) were larger, without a significant increase of the ICNCs$_{pr<106\,μm}$. Therefore, the rime-splintering process probably did not contribute significantly to the increase in ICNC$_{pr<106\,μm}$.





**Figure 7.** (a) Representative examples of ice crystals classified in typical habits observed with HOLIMO between 18:00 UTC and 19:00 UTC on 11 November 2019. The scale bar applies to all panels. (b) Concentrations of ice crystals classified into habits and $ICNC_{pr<106\mu m}$ (black line). (c) Fraction of $ICNC_{pr<106\mu m}$, pristine ice crystals with diameter > 106 μm ($ICNC_{pr<106\mu m}$), aged ice crystals, recirculation particles, and frozen drops concentrations to ICNC. The shaded area shows when HoloBalloon flew out of the cloud. The measurements are averaged over 1 min. The uncertainty for the concentration of ice particles smaller than 106 μm is estimated to ±5% and for the concentration of ice larger crystals to ±15%.



In conclusion, we propose that droplet shattering was mainly responsible for the high peak in $ICNC_{pr<106\,\mu m}$ at 18:10-18:15
       UTC and ice-ice collisions, in particular between recirculation particles, contributed to the peaks in $ICNC_{pr<106\,\mu m}$ between
       18:20-18:55 together with droplet shattering. A comparable $SIP_{high}$ event with $ICNC_{pr<106\,\mu m}$ up to 55 $L^{-1}$ was observed on 12
       November 2019. On this day, columns having formed in higher part of the cloud collided with drizzle drops during sedimen-
       tation, thereby initiating their freezing and splinters production via the droplet shattering mechanism as described in Pasquier
et al. (2022a).

### 3.3    Seeder-feeder event on 1 April 2020

On 1 April 2020, a warm front passed over Ny-Ålesund and caused the observed temperature increase of 7 °C in less than
2 hours, the pressure drop from 1009 hPa to 994 hPa, the wind direction change from southeasterly to northwesterly and the
increase in wind speed at the surface (Fig. 3a,b). Warm air overrunning produced a thickening cirrus cloud, which initially
formed at 7000 m and then continued to deepen into an altostratus cloud (Fig. 8a). The temperature above ∼4500 m a.s.l.
was below -38 °C and thus, the ice crystals formed by homogeneous and/or heterogeneous nucleation in the cirrus/altostratus
cloud. The radar reflectivity signal indicates that ice crystals were sedimenting to about 3000 m a.s.l., where a region of lower
reflectivity suggests their partial sublimation (Fig. 8a). This is in agreement with the relative humidity with respect to ice below
100% measured by the radiosonde above 2500 m a.s.l. (Fig. 4). A low-level cloud formed at around 09:00 UTC with cloud top
height rising from 1000 to 1500 m a.s.l. during the day. This cloud is characterized by regions of higher reflectivity, indicating
the presence of larger ice crystals. Additionally, an increase in reflectivity is visible between 1500 m and 2000 m a.s.l. from
12:00 to 14:00 UTC shows that the layer is saturated with respect to ice, allowing the ice particles to grow, and suggests the
presence of an embedded supercooled liquid layer. This layer can also be seen in the cloud base measured by the ceilometer
when the signal is not attenuated by precipitation.
The CDNCs measured by HOLIMO was generally below 1 $cm^{-3}$ except at 13:10 UTC or between 13:45 and 14:15 UTC,
when increases in CDNC were observed (Fig. 8b). These comparatively large CDNCs (> 10 $cm^{-3}$) are observed when HoloBal-
loon was in the transition region from low to high radar reflectivity (i.e. in the embedded supercooled liquid layer). It suggests
that in this region water saturation was sustained and promoted the formation and growth of cloud droplets, while below, the
environment was subsaturated with respect to water and the cloud droplets were evaporating.
In the low-level cloud, the ICNC amounted up to 55 $L^{-1}$ because of the contribution from crystals sedimenting from the seeder
cloud (Fig. 8). The ice crystal habits were composed of pristine plates and columns together with aged particles (Fig. 9a). The
large aged ice crystals likely originated from the seeder cloud aloft and experienced collisions with cloud particles during
their sedimentation. In contrast, the small columns and plates observed (Fig. 9a) must have formed close to the measurement
location due to their small size and pristine nature. At temperatures below -22°C as experienced above 600 m, supersaturation
relative to ice determines whether ice crystals grow to plates or columns (Nakaya, 1954). The columns therefore originated
from regions with higher supersaturation (likely in the embedded supercooled liquid layer) and plates from region of lower
supersaturation with respect to ice. Indeed, peaks in the concentrations of columns at 13:10 and 14:00 UTC (Fig. 9b) coincide
with the increases in CDNC (Fig. 8b).



**Figure 8.** Overview of the cloud properties on 1 April 2020. (a) Cloud radar reflectivity (color), HoloBalloon measurement height (black line), cloud base height measured by the ceilometer (black dots), and temperatures at the corresponding altitudes measured by the radiosounding at 17:00 UTC. (b) Cloud droplet size distributions (color shading) and total CDNC (black line). .The uncertainty in the concentration of cloud and drizzle is estimated to be ±6%. (c) Ice crystal size distributions (color shading) and total ICNC (black line) measured by HOLIMO averaged over 1 min. The uncertainty for the concentration of ice particles smaller than 106 μm is estimated to ±5% and for the concentration of larger ice crystals to ±15%.



**(a)**

Pristine columns and plates          Aged particles

100μm

Time (UTC)

**Figure 9.** (a) Representative examples of ice crystals classified in typical habits observed with HOLIMO on 1 April 2020. Ice crystals with indication of broken features are highlighted with blue frames. The scale bar is representative for both panels. (b) The concentration of the ice crystals by habit and $ICNC_{pr<106\mu m}$ (black line) between 12:20 UTC and 14:40 UTC (bottom) on 1 April 2020 are shown. The uncertainty for the concentration of ice particles smaller than 100 μm is estimated to ±5% and for the concentration of larger ice crystals to ±15%.

As the $INPC_{HB}$ ($8 \cdot 10^{-2}$ $L^{-1}$) was two to three orders of magnitude lower than the $ICNC_{pr<106\ \mu m}$ (15 $L^{-1}$) (Fig. 3c), SIP
processes were active. Again, we use the ice crystal habits together with the environmental conditions prevailing in this cloud to evaluate the likely SIP processes contributing to $ICNC_{pr<106\ \mu m}$. Rimed particles were observed and the concentration of small droplets may have been sufficient in some regions of the low-level cloud (13:10 UTC and 13:45-14:15 UTC) to trigger the rime-splintering mechanism. However, the observed temperature (-24 ° to -18°C) was far below the temperature range of rime splintering (-8°C to -3°C). Furthermore, no large droplets necessary for the droplet shattering process were observed. Therefore,





the rime-splintering and the droplet shattering processes are unlikely to have played a significant role as SIP mechanisms in the observed cloud. On the contrary, some ice crystals showed broken features, as highlighted by the blue frames in Figure 9a. As the ICNCs were large (up to 55 $L^{-1}$) collisions between ice crystals have likely occurred. In addition, ice-ice collisions is believed to be most efficient at colder temperature (Takahashi et al., 1995) such as observed on this day. Therefore, we deduce that the ice-ice collisions were again the most likely active SIP mechanism in the low-level feeder cloud.

## 4   Environmental conditions favorable for SIP

During the six days of observations performed with HoloBalloon during the NASCENT campaign, 2252 measurements of 30 s intervals were taken in-cloud, corresponding to a total of 18.7 hours and a volume of 6425 L. Out of these measurements, $SIP_{all}$ (representing all measurements with $ICNC_{pr<106\,\mu m} > 0.3\ L^{-1}$) was present during 40% of the measurements. When dividing by the intensity of the SIP, $SIP_{low}$, $SIP_{mod}$ and $SIP_{high}$ occurred 20.5%, 16%, and 3.5% of the time, respectively (Fig. 10).

As described in Section 1, several environmental conditions (e.g., cloud droplet concentration and size, ice crystal size and habit, and temperature) influence the occurrence of SIP. Using the assumption that pristine ice crystals smaller than 106 μm are associated with their environment of origin, we can relate SIP to the environmental conditions prevailing at the measurement location. The role of the different hydrometeor types and temperatures for the occurrence of SIP observed on the six days of measurements in MPCs is discussed below.

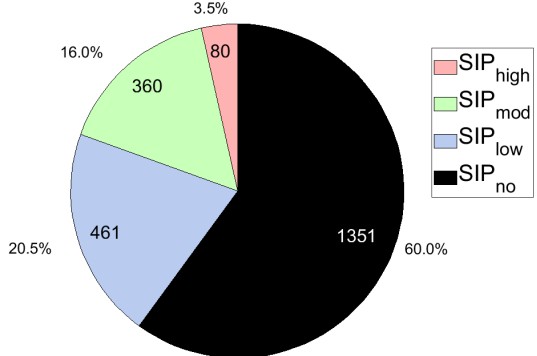

**Figure 10.** Frequency of occurrence of $SIP_{no}$ ($ICNC_{pr<106\,\mu m} < 0.3\ L^{-1}$), $SIP_{low}$ ($0.3\ L^{-1} < ICNC_{pr<106\,\mu m} < 1\ L^{-1}$), $SIP_{mod}$ ($1\ L^{-1} < ICNC_{pr<106\,\mu m}$ $< 10\ L^{-1}$), and $SIP_{high}$ ($10\ L^{-1} < ICNC_{pr<106\,\mu m}$). The numbers refer to the number of 30 s intervals observed within each SIP class.

### 4.1   Role of the hydrometeor types for SIP

The comparison between $ICNC_{pr<106\,\mu m}$ representative of SIP and the concentrations of cloud droplets (diameter < 64 μm), drizzle drops (diameter > 64 μm), frozen drops, and ice crystals help to understand their relationship to SIP. The analysis of the influence of ice crystals on SIP is delicate because it is possible that the larger ice crystals are secondary ice crystals having grown to larger sizes than the threshold used (106 μm). To overcome this issue, we discuss only the connection between SIP
and ice crystals larger than 327 μm, and refer to these as snow crystals.









**Figure 11.** (a) $ICNC_{pr<106\,\mu m}$, (b) CDNC and drizzle drop number concentrations (DDNC), (c) frozen drop number concentrations, and (d) snow crystals number concentrations retrieved with HOLIMO averaged over 30 s. The uncertainty for the concentration of cloud droplets is estimated to $\pm 6\%$, for the concentration of ice particles smaller than 100 μm to $\pm 5\%$ and for the concentration of snow crystals and frozen drops to $\pm 15\%$. (e) Temperature derived from the radiosondes at the HoloBalloon location. The breaks on the time axis separate measurement flights. The black dashed lines in panel (a) and (c) denote the $SIP_{mod}$ (1 L$^{-1}$) and $SIP_{high}$ (10 L$^{-1}$) limits. The white regions show the occurrence of SIP, whereas the grey shaded regions show no SIP.

Snow crystals seem to follow the same trend as $ICNC_{pr<106\,\mu m}$ (Fig. 11a,d) and the correlation coefficient between the concentrations of snow crystals and $ICNC_{pr<106\,\mu m}$ amounts to 0.4. This demonstrates the obvious connection between snow crystals and SIP, i.e. primary ice is needed in order for SIP to be initiated. On the contrary, no obvious connection between $ICNC_{pr<106\,\mu m}$ and cloud droplet is observed (correlation coefficient of 0.01). Indeed, the highest CDNCs prevail on 10 Novem-
ber 2019, when no evidence for SIP is observed and the CDNCs are mostly below 5 cm$^{-3}$ during the prevalence of $SIP_{mod}$ and $SIP_{high}$ events on 11 and 12 November 2019 (Fig. 11a,b). Drizzle drops are always observed during SIP occurrence, except on 1 April 2020, when only snow crystals are observed (Fig. 11a,b,d). This suggests that on 1 April 2020, the presence of snow crystals alone was sufficient for the occurrence of SIP, likely via the ice-ice collision process as discussed in Section 3.3. During the first flight on 11 November 2019, the highest drizzle drop number concentrations (up to 20 L$^{-1}$) were measured, but
no SIP was observed. The reason is likely that there were not enough snow crystals colliding with the drizzle drops, thereby not initiating their freezing causing a lack of SIP via the droplet shattering process. In fact, no frozen drops were observed on this day. This indicates that freezing of drizzle drops via immersion or contact freezing with an INP is not sufficient to trigger droplet shattering at the temperature experienced (-8°C to -2°C), but the presence of snow crystals is needed to initiate their freezing. Indeed, frozen drops are observed during 41.5% of $SIP_{all}$ and 87.5% of the $SIP_{high}$ events (Table 1).
To quantify the importance of different hydrometeor types for SIP, we calculate an occurrence enhancement factor (OEF) relative to $SIP_{no}$ for all the SIP classes and for the hydrometeor types: cloud droplets, drizzle drops, frozen drops, and snow crystals. First, the frequency of occurrence of a hydrometeor type during each SIP class ($F_{SIP_{class}}$) and the frequency of occurrence of a hydrometeor type during when no SIP is observed ($F_{SIP_{no}}$) were calculated. Then, the OEF for every hydrometeor type and SIP class ($OEF_{SIP_{class}}$) was derived as follows:

$$OEF_{SIP_{class}} = \frac{F_{SIP_{class}}}{F_{SIP_{no}}} \tag{1}$$

An OEF greater than unity signifies that the hydrometeor type is more frequently present during SIP than during $SIP_{no}$ and thus hints at a possible connection between the hydrometeor type and the occurrence of SIP.

During the presence of snow crystals, the frequency of occurrence of $SIP_{all}$ compared to $SIP_{no}$ is enhanced by a factor of 2.7, and $SIP_{high}$ by a factor of 2.9 (Table 1). This further demonstrates that the production of ice crystals prior to SIP is required.
The influence of a high concentration of cloud droplets on SIP was identified by using a threshold of CDNC > 5 cm$^{-3}$, which represents the mean CDNC over the six measurement days. The OEF of cloud droplets is below 1 for all SIP classes except $SIP_{high}$, for which it is slightly increased to 1.33 (Table 1). This signifies that the occurrence of SIP is reduced compared to





$SIP_{no}$ when the concentration of cloud droplets was higher than 5 cm$^{-3}$ and indicates that concentrations of cloud droplets

exceeding 5 cm$^{-3}$ were not necessary for SIP occurrence in the measurements presented. In contrast, the occurrences of all SIP

classes are enhanced when drizzle drops are present, suggesting an influence of the droplet shattering mechanism. Finally, the

occurrence of SIP is enhanced by a factor between 3 and 10 compared to $SIP_{no}$ when frozen drops are observed (Table 1).

This large enhancement is also consistent with a dominant role of the droplet shattering mechanism, especially for $SIP_{mod}$ and

$SIP_{high}$.

Previous studies have linked the presence of drizzle drops to the occurrence of SIP in tropical and midlatitude convective

clouds (e.g., Lawson et al., 2015, 2017; Keppas et al., 2017). In convective clouds with a warm cloud base, the formation

of drizzle drops occurs by collision-coalescence in updraft cores that extend over a large portion of the troposphere (Lawson

et al., 2017). In other cases, the drizzle drops responsible for the initiation of droplet shattering close to the melting layer were

suggested to originate from melted ice crystals recirculating through the melting layer within updrafts (Korolev et al., 2020;

Lauber et al., 2021). Here, we propose that the formation of large drizzle drops, which are related to SIP, is determined by

the low CCN concentration prevailing in the clean Arctic environment, together with the sufficiently high updraft speeds as

observed in cloud containing drizzle drops during NASCENT. A connection between drizzle drops and ice crystal formation

was already proposed by Rangno and Hobbs (2001) and Lance et al. (2011). However, they did not relate the formation of the

ice crystals to SIP via the droplet shattering mechanisms.

In summary, no connection was found between the concentration of cloud droplets exceeding 5 cm$^{-3}$ and SIP. On the contrary,

a strong relationship exists between drizzle drops and SIP, with the prerequisite that sufficient snow crystals are present to

initiate their freezing upon collision and activate the droplet shattering process. Moreover, snow crystals can be sufficient for

triggering SIP via ice-ice collisions.

**Table 1.** Frequency of occurrence and OEF of the hydrometeor types cloud droplets (with concentrations larger than 5 cm$^{-3}$), drizzle drops, frozen drops, and snow crystals during all measurements ($N_{all}$), $SIP_{all}$, $SIP_{low}$, $SIP_{mod}$, and $SIP_{high}$. Bold font signifies OEF values larger than 1, i.e. enhancements.

|  |  | $N_{all}$ | $SIP_{no}$ | $SIP_{all}$ | $SIP_{low}$ | $SIP_{mod}$ | $SIP_{high}$ |
|---|---|---|---|---|---|---|---|
| Cloud droplets | F (%) | 32.5 | 35.5 | 28 | 28.5 | 22.5 | 47.5 |
|  | OEF |  |  | 0.78 | 0.8 | 0.63 | **1.33** |
| Drizzle drops | F (%) | 58.5 | 53.5 | 65.5 | 69.5 | 56.5 | 85 |
|  | OEF |  |  | **1.22** | **1.3** | **1.05** | **1.59** |
| Frozen drops | F (%) | 22 | 8.5 | 41.5 | 32 | 77.5 | 87.5 |
|  | OEF |  |  | **4.95** | **3.78** | **9.2** | **10.37** |
| Snow crystals | F (%) | 58 | 35 | 93 | 87 | 99 | 100 |
|  | OEF |  |  | **2.7** | **2.5** | **2.8** | **2.9** |


## 4.2 Temperature

During the six days of MPC observations, measurements covered temperatures between -24 °C and -1 °C, albeit with very few

measurements between -14 °C and -10 °C (Fig. 12c,d). Between -8 °C and -2 °C, evidence of SIP was observed between 55% and 75% of the time (Fig. 12c). Meanwhile, at temperatures below -18 °C, evidence of SIP was almost always observed, with 96% of the measurements involving SIP (Fig. 12c). However, the measurements obtained at these low temperatures originate solely from 1 April 2020 (Fig. 12d) and are related to the ice-ice collision process, as discussed in Section 3.3. It should also be noted that the large number of measurements without SIP at -16 °C occurred during the cloud case on 10 November 2019

(Fig. 12d) , when ice formation was limited by the INPC, as discussed in Section 3.1 (see also the temperature evolution during the flights in Figure 11e).

In addition to the frequency of occurrence of SIP, the number of secondary ice crystals produced determine the impact of SIP. The distribution of the fraction of $ICNC_{pr<106\,\mu m}$ to total ICNC as a function of temperature and $ICNC_{pr<106\,\mu m}$ (Fig. 12b) gives information on the number of ice crystals produced by SIP at each temperature. The highest $ICNC_{pr<106\,\mu m}$ were observed

at temperatures between -7 °C and -2 °C, with concentrations exceeding 50 $L^{-1}$ (i.e., in the $SIP_{high}$ class) between -5 °C and -3 °C (Fig. 12b). Measurements performed on 11 and 12 November 2019 are responsible for this $SIP_{high}$ event (Fig. 12d) and are mainly caused by the droplet shattering and the ice-ice collision processes (as discussed in Section 3.2 and Pasquier et al. (2022a)). Moderate to high $ICNC_{pr<106\,\mu m}$ ($SIP_{mod}$ and $SIP_{high}$ classes) were also observed at temperatures between -24 °C and -16 °C on 1 April 2020 (Fig. 12b,d). Note that the warmer temperature range (-7 °C and -2 °C) overlaps with the rime-

splintering process. However, since the other criteria for the rime-splintering process (i.e., rimed ice crystals and a sufficient concentrations of cloud droplets with diameters smaller than 12 μm) were not met during the measurements with SIP, the contribution of the rime-splintering process is assumed to be negligible.

The concentrations of small ice crystals are higher (Fig. 12b), but the proportion of measurements with SIP occurrence (Fig. 12c) was lower on 11 and 12 November 2019 between -7 °C and -2 °C, compared to measurements obtained on 1 April

2020 between -24 °C and -18 °C. Thus, the droplet shattering processes found to be active at the warmer temperatures on 11 and 12 November seems to be less frequently active but to create more splinters than the ice-ice collision process found to be active at the colder temperatures on 1 April 2020. This would be in agreement with laboratory studies showing that a large number of splinters (>10) can be produced from the freezing of a single drop (Lauber et al., 2018; Korolev and Leisner, 2020) Note however that one measurement flight at lower temperature is not sufficient to draw a conclusive statement about

the number splinters produced at these temperatures.

To conclude, SIP occurred over the entire temperature range where measurements were performed, with the highest concentrations of ice crystals smaller than 106 μm (>50 $L^{-1}$) observed between -3° and -5 °C caused mainly by the droplet shattering process and the highest percentage of the measurements with SIP between -18° and -24 °C caused by the ice-ice collision mechanism. This denotes the importance of the droplet shattering and ice-ice collision mechanisms over a large temperature

range and highlights the necessity to include these processes over a larger temperature range in numerical weather and climate models.





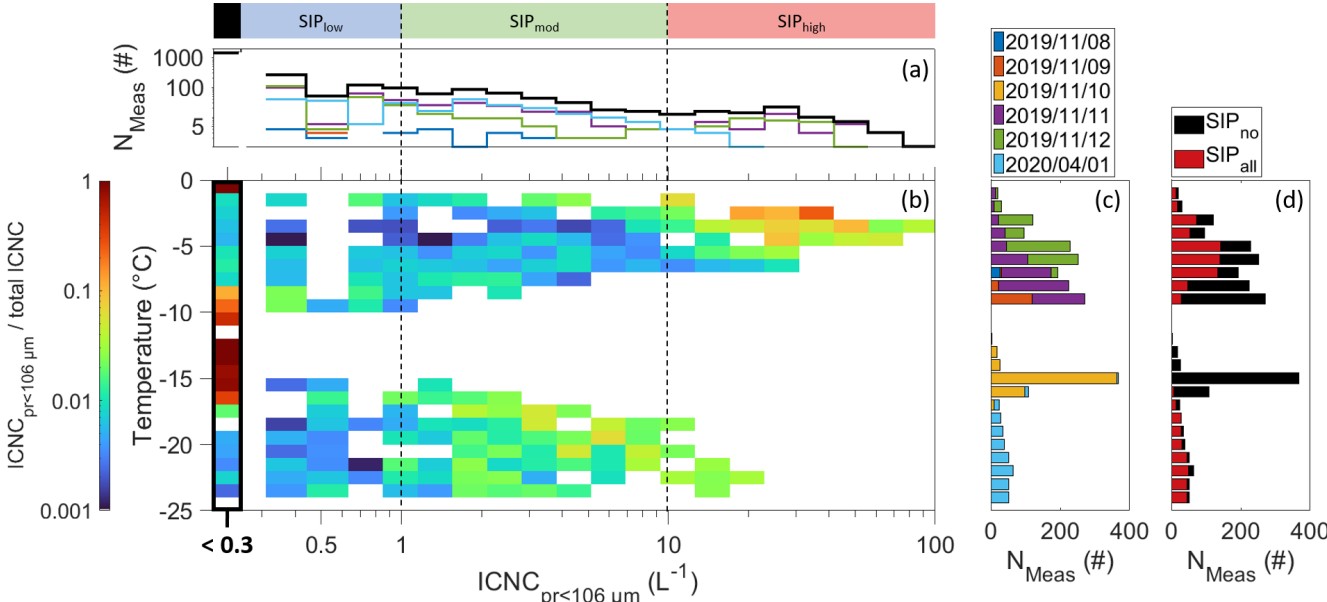

**Figure 12.** (a) Number of measurements for each ICNC$_{pr<106\mu m}$ bin (note the log scale) for each day of measurements (color lines) and all measurements (black line). The ICNC$_{pr<106\mu m}$ regions defined as SIP$_{low}$, SIP$_{mod}$, SIP$_{high}$ are shown on top and SIP$_{no}$ is represented with a black box. (b) ICNC$_{pr<106\mu m}$ fraction from total ICNC for each temperature bin of 1 °C (color shading) and ICNC$_{pr<106\mu m}$ bin. The frequency of ICNC$_{pr<106\mu m}$ < 0.3 L$^{-1}$ to ICNC (SIP$_{no}$ class conditions) is highlighted by the thick black frame. A concentration of 0.3 L$^{-1}$ was used for the calculation of ICNC$_{pr<106\mu m}$ to total ICNC when no ice crystal was measured in the 30 s interval. (c) Number of measurements (N$_{meas}$) per temperature bin (1 °C) for measurements with SIP (red bars), and for measurements with SIP$_{no}$ (black bars). (d) Number of measurements (N$_{meas}$) per temperature bin for each day of measurements (colored lines). The data were averaged over 30 s for the analysis.

## 5   Conclusions

In this paper, the microphysical properties of Arctic MPCs measured during the NASCENT campaign with the tethered balloon system HoloBalloon during five consecutive days from 8 to 12 November 2019 and on 1 April 2020, together with

ground-based INP and remote sensing measurements as well as radiosonde profiling are discussed. An emphasis is placed on the formation of ice crystals, especially on the occurrence of SIP, and on the environmental conditions favorable for SIP. We used the concentration of small pristine ice crystals (ICNC$_{pr<106\ \mu m}$) to identify SIP occurring in the 60 to 120 s preceding the measurements. The key findings are summarized as follows:

- SIP regions were identified in 40% of the in-cloud measurements. In one probed MPC on 10 November 2019, ice crystal formation was limited by the concentration of aerosols acting as INPs at -17 °C. In two other MPCs on 11 and 12 November 2019, the ICNC$_{pr<106\ \mu m}$ suddenly increased from below 1 L$^{-1}$ (SIP$_{low}$) to more than 50 L$^{-1}$ (SIP$_{high}$) due to the droplet shattering mechanism, which most likely generated a positive SIP feedback loop by creating splinters causing the freezing of additional



droplets, creating splinters again. Finally, in two MPCs on 11 November 2019 and on 1 April 2020, the ice-ice collision mechanism was proposed to be responsible for moderate to high SIP ($ICNC_{pr<106\,\mu m}$ up to 15 L$^{-1}$).

- Drizzle drops were found to be favorable for the occurrence of SIP, as the frequency of SIP was enhanced in the presence of drizzle drops. Moreover, the frequency of occurrence of frozen drops was enhanced by a factor of 5 during SIP events (Table 1), whereby frozen drops were measured in 87.5% of the $SIP_{high}$ observations. Thus, freezing of drizzle drops was strongly favorable for SIP, which indicates a large contribution from the droplet shattering mechanism. We relate the presence of drizzle drops itself to the strong updrafts and low CCN concentrations observed in the clean Arctic environment.

- SIP cloud regions were observed over a large temperature range (-24 °C to -1 °C). The highest concentrations of secondary ice crystals were measured between -5 °C and -3 °C (>50 L$^{-1}$, Fig. 12b) and related mainly to the droplet shattering mechanism (Section 3.2), while the highest proportion of the measurements showed the occurrence of SIP between -24 °C and -18 °C (up to 95%, Fig. 12c) in one MPC related to the ice-ice collision mechanism (Section 3.3). This emphasizes the need to include SIP parametrizations for this two processes over a large temperatures range in numerical weather prediction models, which generally only include a parametrization for the rime-splintering process active at temperatures between -8 °C and -3 °C.

Overall, this study observed a large variety of microphysical properties of Arctic MPCs during the six days of measurements including two SIP mechanisms and the conditions favorable for these SIP mechanisms were discussed. However, further field and laboratory studies are required to better constrain the environmental conditions favorable for SIP in order to develop robust SIP parametrizations for numerical weather prediction models. In particular, field studies should characterize in-cloud INPC up to high sub-freezing temperatures (>10 °C) to accurately constrain the SIP rate. Furthermore, we especially recommend to include the presence of drizzle drops and their collision frequency with ice to estimate the contribution from the droplet shattering mechanism, which was shown to play an important role for ice crystal formation in the observed Arctic MPC. Finally, we propose to extend the SIP parametrizations to all sub-freezing temperatures, as SIP was observed down to -24 °C in one sampled Arctic MPC.

*Code and data availability.* The cloud microphysical and aerosol datasets as well as the scripts to reproduce the figures will be available on Zenodo (https://zenodo.org/). Radiosonde and surface weather data are available in PANGAEA (https://www.pangaea.de/) (Maturilli, 2020d, a, c, b).



## Appendix A: Auxiliary parameters

### A1  Potential temperature and wind profile

The potential temperature and wind profiles observed from the radiosondes on the six days of measurements suggest well-
mixed boundary layers and no strongly decoupled cloud is observed.

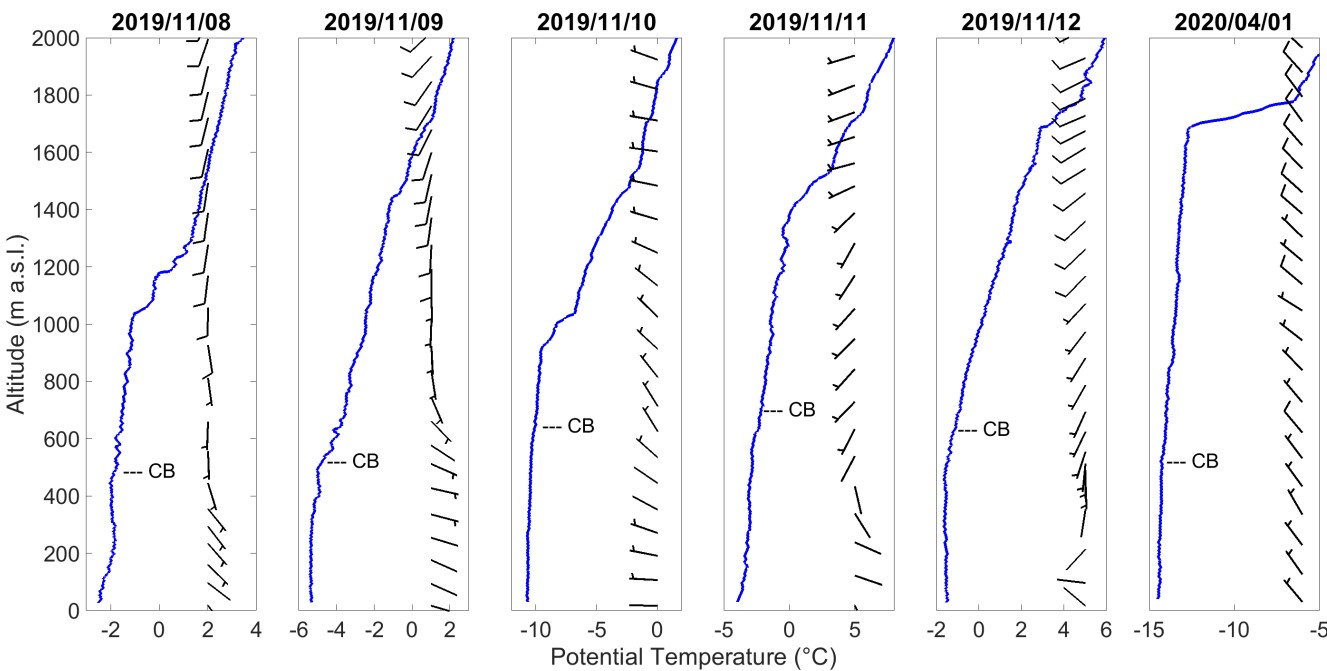

**Figure A1.** Potential temperature and wind speed and direction measured by the radiosonde launched at 11:00 UTC or 17:00 UTC on the six
days of measurements. The mean cloud base (CB) measured with the ceilometer is labeled.

### A2  Cloud top and HoloBalloon temperature and relative humidity determination from radiosonde measurements

The temperature profile from the radiosondes was used to determine the ambient temperature at HoloBalloon's measurement
location and the cloud top temperature. If several radiosondes were launched during a day, the temperature profile between two
launches was linearly interpolated from the two closest profiles. If only the daily radiosonde was launched, the temperature
profile was used for the whole day. The same method was applied for the relative humidity. The cloud top altitude was deter-
mined from the first altitude where the cloud radar does not measure the reflectivity and a running mean over 5 minutes was
used to smooth high temporal variability in cloud top height. From this altitude the temperature at cloud top was derived.





## A3 Updraft wind speed estimate

As the Doppler velocity is the sum of the fall velocity of cloud particles and updraft/downdraft, the largest Doppler velocities
within a measured Doppler spectrum can be used as approximation for the updraft velocities experienced by the smallest cloud
particles (Shupe et al., 2008) . We use a similar approach as in Ramelli et al. (2021) to estimate the updraft velocity from
the maximum Doppler velocity derived from the Doppler spectra as shown in Figure A2. First, a running mean was used to
smooth the Doppler spectra. If the difference between $Z_{max}$ and $Z_{min}$ exceeded 20 dBZ, the maximum Doppler velocity $v_{max}$
was derived as follows:

$v_{max} = $ maximal Doppler velocity where $Z \geqq (Z_{min} + 0.2 \cdot (Z_{max} - Z_{min}))$ (A1)

where $Z_{max}$ and $Z_{min}$ are the maximum and minimum radar reflectivity. If the difference between $Z_{min}$ and $Z_{min}$ was lower
than 20 dBZ, $v_{max}$ was derived at -47 dBZ to avoid the selection of noise in Doppler spectra with low amplitude. The threshold
of -47 dBZ was chosen because it is the lowest reflectivity that was typically above the noise level. A positive (negative)
Doppler velocity indicates downdraft (updraft). Note that in the absence of small cloud particles, the updraft may be strongly
underestimated by this method.

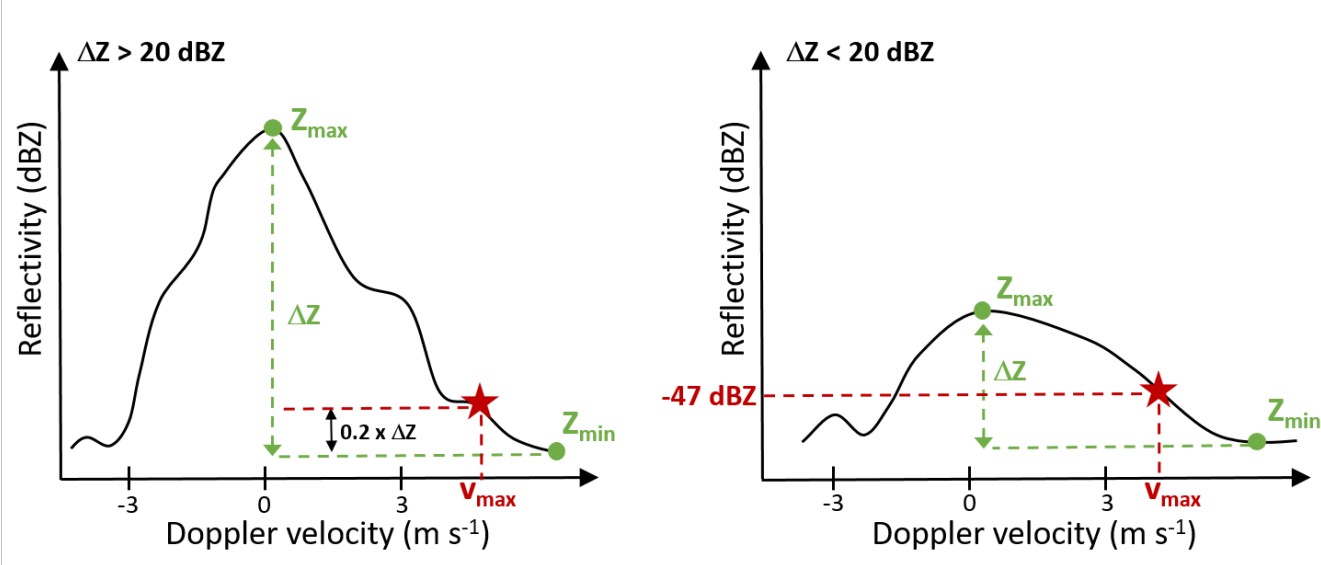

**Figure A2.** Schematic of the derivation of the maximum Doppler velocity $v_{max}$ (red star) from the Doppler spectra. $Z_{min}$ and $Z_{max}$ (green
dots) are the minimum and maximum radar reflectivity (see the text for more details).

*Author contributions.* JTP analyzed the cloud observational data and prepared the figures of the paper. FR, JH, ROD, AL, JW, and UL helped
in analyzing and interpreting the observational data. JTP, JW, ROD, TC, JH performed the HoloBalloon measurements. JW performed the



INP measurements. RG processed the remote sensing data and helped in interpreting the remote sensing observations. MM was responsible for the radiosonde launches during the NASCENT campaign. JTP prepared the manuscript with contributions from all authors.

*Competing interests.* The authors declare that they have no conflict of interest.

*Acknowledgements.* This project has received funding from the European Union's Horizon 2020 research and innovation programme under grant agreement No 821205 (FORCeS), from the Swiss Polar Institute (Exploratory Grants 2018), and from the Swiss National Science Foundation (SNSF) (grant no. 200021_175824). RG and MM gratefully acknowledge the funding by the Deutsche Forschungsgemeinschaft (DFG, German Research Foundation) – Project-ID 268020496 – TRR 172, within the Transregional Collaborative Research Center "ArctiC

Amplification: Climate Relevant Atmospheric and SurfaCe Processes, and Feedback Mechanisms (AC)3". ROD and TC gratefully acknowledge the funding by the European Research Council (ERC) through Grant StG758005. We thank Prof. Alexei Korolev for the fruitful scientific discussions. We thank Michael Roesch for his help in the installation of the setup for the campaign. We would also like to particularly thank Roland Neuber and Paul Zieger for their support and advice during the organisation of the campaign. We thank all those involved in the field work associated with NASCENT, particularly the AWIPEV and Norwegian Polar Institute Sverdrup stations staff.



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
