# Peer review of "Conditions favorable for secondary ice production in Arctic mixed-phase clouds"

_Atmospheric Chemistry and Physics, 2022_

## Referee Comment (RC1)

**Review for:**

**Conditions favorable for secondary ice production in Arctic mixed-phase clouds**

*by Julie T. Pasquier*

**Summary:**

Pasquier et al. investigate the conditions that favor secondary ice production (SIP) in Arctic clouds, observed during NASCENT campaign, and examine the possible underlying mechanisms. For their investigations they use in-situ cloud microphysical measurements derived by a holographic cloud imager on a tethered balloon system and ground-based remote-sensing and ice nucleating particle (INP) measurements. Their analysis focuses on a six-day period which revealed the occurrence of SIP for about 40% of the time, while very high SIP events (with ICNCs > 10 L$^{-1}$) were identified in 3.5% of the analyzed data. The highest SIP efficiency was found at temperatures between -3$^{\circ}$C and -5$^{\circ}$C; interestingly, this was attributed to the drop-shattering mechanism rather than to the Hallett-Mossop process, which has been thought to be the dominant SIP mechanism at these temperatures. Ice-ice collisions is also identified as the second most important SIP mechanism, especially at colder temperatures down to -24°C.

This is a very well-written paper and a very interesting study. Models fail to reproduce the micro- and macro- physical structure of Arctic clouds and the description of ice microphysical processes has long been known to be a main contributor to these errors. Also while SIP has been hypothesized to be responsible for the enhanced cloud ice number concentrations often observed in the pristine Arctic environment, where INP availability is limited, the exact SIP mechanisms remain unknown. In general, there are very few in-cloud ICNC datasets from the Arctic and also most of them are not combined with INP measurements, which makes it difficult to quantify the influence of primary ice production versus SIP. This highlights the importance of the present study and the analyzed datasets for understanding ice production in Arctic clouds. For this reason, I recommend this paper for publication after the comments below have been addressed.

**Comments:**

**Line 63-64:** maybe discuss a bit how ice shape is expected to influence SIP (with references)

**Line 105:** since measurements were collected during four months in total, why only such a small sample of six days is presented here? Please explain

**Line 111-114:** are particles below 25 micro re-examined manually or are they treated as droplets in the analysis? In the case they are treated as droplets, can you estimate the magnitude of SIP underestimation? Fragments generated by drop-shattering can about 10 micro (Phillips et al. 2018), while these can be even smaller for the other two SIP processes. The same question concerns misclassified ice crystals with circular shape.

**Line 114-115:** could you provide more details on the criteria (characteristics) used to

classify to ice particles as recirculated or aged? How do you separate these two categories?

**Lines 125-126:** what do you mean 'minimized'? Could you provide an estimate for how frequently shattering occurs? You should use the inter-arrival time algorithm (Korolev and Field 2015) to identify shattering artifacts and exclude them from the analysis.

**Lines 153-154:** There is something I don't understand about this method. Why pristine ice crystals with size < 106 micro cannot be newly-formed primary ice crystals? Why these should solely be associated with SIP? Please explain.

**Lines 180-181:** provide reference

**Lines 193-195:** I am not convinced that the profiles derived on 8/11 and 11/11 are well-mixed. A $\Theta$-gradient of $0.5^{\circ}C$ is often used as criterion for decoupling (Sotiropoulou et al. 2014, Gierens et al. 2020). Please use one of the proposed methods in the literature to ensure cloud-surface coupling.

**Lines 238-239:** you do not provide any information on updraft velocity for November 10 to support this statement. I suggest to provide a time-height cross-section for this parameter (at least in the appendix or as supplementary material).

**Section 3.2:** While this section focuses on the investigation of the high SIP event in the afternoon of 11/11, it is worth including a short discussion for the possible drivers of the weaker SIP before 18:00.

**Lines 332-333:** However, peaks in the columnar ice concentrations before 13:00, which are of similar magnitude as the one observed round 13:00-13:155, are not associated with CDNCs increases. What is the reason behind their formation?

**Line 333:** There seems to be an almost constant white shading at sizes between 7-8 micro in both Figures 6 and 8. Is this some kind of artifact?

**Line 342:** It worths discussing here a bit more about the ice-ice collision process. Is it the same mechanism here as proposed by Geogakaki et al. (2022) for seeder-feeder events? They suggest that ice particles falling from the upper cloud collide with ice particles within the lower mixed-phase layer, resulting in mechanical break-up. And if this the case, why the process does not take place earlier, since the seeding-feeding system is observed from approximately 12:00 to 14:00?

**Line 344:** please provide the reference of Georgakaki et al. (2022) here, who reached the same conclusion about seeder-feeder cases.

**Line 344:** do you exclude the possible contribution of sublimation break-up? Deshmukh et al. (2022) suggest that as precipitation particles falling from the seeder cloud into a sub-saturated environment may experience sublimation break-up. Then, as the new fragments enter the saturated conditions of the feeder cloud, they can further grow through vapor deposition and enhance ICNCs (see their schematic in Fig. 14).

**Line 361:** Could the correlation between ice-snow concentrations be due to the fact that SIP particles will eventually grow to snowflakes? This means that increases in snow

concentration would follow the increases in the number smaller ice particle.

**Line 367-368:** Again, the contribution of sublimation break-up should be investigated here (see comment above)

**Line 385:** How sensitive are the results to the choice of this CDNC threshold?

**Lines 386-387:** This comment concerns the small OEF for cloud droplets. The presence of cloud drops is expected to be important for SIP within the Hallet-Mossop temperature zone, but not outside of it. If you calculate the OEF only for the H-M zone, does the factor changes significantly?

**Lines 400:** updraft speeds are hardly discussed in this manuscript. I suggest to add a figure for this variable and discuss this more thoroughly in relevance to SIP occurrence and drizzle formation

**Line 433:** Please add the reference of Luke et al (2021) here. They were the first to show that high SIP events are associated with the presence of large drops in Arctic clouds. It is worth trying to relate your analysis to their findings, derived with remote-sensing methods.

**Line 450:** Could you infer a minimum INPC that is necessary to initiate SIP from your measurements? Or at least an INPC level below SIP is never expected to occur.

**Line 461:** This is a statement that is not clearly supported by the analyzed data. The connection of updraft velocities, CCN and drizzle concentrations should be shown explicitly in the figures.

**Lines 465-466:** if the contribution of sublimation break-up cannot be excluded with the existing data, maybe the possibility of having more mechanisms activated should be addressed here.

**REFERENCES:**

Deshmukh, A., Phillips, V. T. J., Bansemer, A., Patade, S., & Waman, D. (2022). New Empirical Formulation for the Sublimational Breakup of Graupel and Dendritic Snow, *Journal of the Atmospheric Sciences*, *79*(1), 317-336.

Gierens, R., Kneifel, S., Shupe, M. D., Ebell, K., Maturilli, M., and Löhnert, U.: Low-level mixed-phase clouds in a complex Arctic environment, Atmos. Chem. Phys., 20, 3459–3481, https://doi.org/10.5194/acp-20-3459-2020, 2020.

Korolev, A. and Field, P. R.: Assessment of the performance of the inter-arrival time algorithm to identify ice shattering artifacts in cloud particle probe measurements, Atmos. Meas. Tech., 8, 761–777, https://doi.org/10.5194/amt-8-761-2015, 2015.

Luke, E.P., F. Yang, P. Kollias, A.M. Vogelmann, and M.,Maahn, 2021: New insights into ice multiplication using remote-sensing observations of slightly supercooled mixed-phase clouds in the Arctic. *Proceedings of the National Academy of Sciences,* **118**. doi: 10.1073/pnas.2021387118

Phillips, V. T. J., Patade, S., Gutierrez, J., & Bansemer, A. (2018). Secondary Ice Production by Fragmentation of Freezing Drops: Formulation and Theory, *Journal of the Atmospheric Sciences*, *75*(9), 3031-3070.

Sotiropoulou, G., Sedlar, J., Tjernström, M., Shupe, M. D., Brooks, I. M., and Persson, P. O. G.: The thermodynamic structure of summer Arctic stratocumulus and the dynamic coupling to the surface, Atmos. Chem. Phys., 14, 12573–12592, https://doi.org/10.5194/acp-14-12573-2014, 2014.

---

## Referee Comment (RC2)

**Review of "Conditions favorable for secondary ice production in Arctic mixed-phase clouds" by Pasquier et al.**

**Overview**

The paper presents an important in-situ observation of secondary ice production in Arctic clouds from a tethered balloon with the help of a high-resolution imaging probe HOLIMO. The HOLIMO provided photographic quality images of cloud particles, which, in the majority of cases, allowed for unambiguous identification of particles habit and their phase. The in-situ observations were complemented by remote sensing observations, which allowed for accurate positioning of the HOLIMO measurements with respect to cloud boundaries. Ground based measurements of INPs provided a significant contribution to the value and quality of the collected data set. One of the major outcomes of this study is lowering the threshold concentration of ice particles separating primary and secondary ice production. This is different from the past observations when SIP was identified as an explosive increase in the number concentration of ice particles exceeding hundreds and thousands per liter. In the present study, the threshold concentration was reduced to the order of 0.1-10 per liter. The paper provides an important contribution to the understanding of the role of SIP in ice formation and, undoubtedly, it should be published in ACP. The paper is well written, and I do not have any significant comments on this work. There are a few minor comments which are worth addressing prior to publication.

**Recommendation**: The paper should be published in ACP after minor revisions

**Comments**

1.  Line 204 and throughout the text: "*drizzle drops (defined with diameter larger than 64 μm)*"
    Following commonly accepted definition (e.g., Glossary of Meteorology), drizzle is defined as drops in the size range 100μm<D<500μm. The authors may consider using the term "supercooled large droplets" (SLD) instead. SLD is defined as droplets with D>50μm at T<0C.

2.  Line 124-126: "*In addition, thanks to the low true air speed of HOLIMO on the tethered balloon system and the adequate tower tips, the shattering of ice crystals in the sample volume is minimized.*"
    Reduction of the sampling speed will undoubtedly reduce the effect of shattering of ice due to decreasing the kinetic energy of particles impact. However, the HOLIMO inlet axis is unlikely to be always perfectly aligned with the local airflow, and particles with anisoaxial trajectories are likely to be frequently present during sampling. This is specifically relevant to the cases with the turbulent environment. Particle fall speed will also contribute to the deviation of the particle trajectories from the inlet flow. Such particles will impact with the HOLIMO inlet walls at a speed of ~ 1-2m/s, and they may get fragmented and contaminate measurements shattering artifacts. Fragmentation of freefalling (1-2m/s) particles on impact with a solid surface of observed by Vardiman (JAS, 1978). In this regard, it would be reasonable to make a disclaimer when talking about the effect of shattering.

3.  Line 158-159: "*Non-pristine crystals cannot have formed from vapor deposition growth, and could originate from breakups during impact with the instrument payload or from rime falling from the tethered balloon.*"
    The first part of this statement should be corrected. For example, non-pristine ice particles can be formed during water vapor deposition growth on polycrystalline frozen droplets or developed after formation dislocation in the ice crystal lattice of pristine particles.

4. Line 290: "*Indeed, particles resembling broken branches were observed (highlighted with the dark brown box in Figure 7a).*"
   Could you elaborate why the images of the boxed particles were identified as broken branches. Particle having such shape could be formed without fragmentation.

5. Line 320: "*The CDNCs measured by HOLIMO was generally below 1 cm-3 except at 13:10 UTC or between 13:45 and 14:15 UTC, when increases in CDNC were observed (Fig. 8b). These comparatively large CDNCs (> 10 cm-3) are observed when HoloBalloon was in the transition region from low to high radar reflectivity (i.e. in the embedded supercooled liquid layer). It suggests that in this region water saturation was sustained and promoted the formation and growth of cloud droplets, while below, the environment was subsaturated with respect to water and the cloud droplets were evaporating.*"
   I have serious concerns regarding the existence of sustainable liquid clouds in the boundary layer with droplets D<40µm and number concentration <1cm-3 (e.g., as shown in Fig.8b). Such clouds have a high level of instability and may exist in a relatively turbulent environment only for a limited time since any vertical motion will result in activation of CCNs or complete evaporation of droplets. Therefore, the interpretation of observations of cloud segments with N<0.1cm-3 as liquid or mixed-phase causes a question of whether the phase of particles with D<40µm was identified correctly, or the images of these particles are a result of some non-recognized artifacts.

   In relation to Fig 8, I would like to add that the rapid increase of the radar reflectivity from -10dBZ to ~10dBZ at approximately 1km altitude is indicative of a presence of a liquid layer there. Such layers usually result in the enhanced growth of seeded from above ice crystals and an increase in radar reflectivity. On the other hand, Arctic clouds are frequently decoupled. I am wondering if the cloud base identified from the ceilometer measurements, as shown in Fig.8a, corresponds to the lower layer. This may result in a perception that the HOLIMO measurements were performed inside a liquid/mixed-phase cloud layer, whereas the measurements in fact were sampled in between cloud layers. Could you also comment on this?

6. Line 341-342: "*On the contrary, some ice crystals showed broken features, as highlighted by the blue frames in Figure 9a. As the ICNCs were large (up to 55 L-1) collisions between ice crystals have likely occurred.*"
   I would be conservative regarding the interpretation of the boxed images in Fig.9a as fragments. Images of ice particles with under- or non-developed corners or branches can be found in Nakaya (1954) or Bentley and Humphreys (1931), respectively. Could you comment or provide strong evidence supporting the fragmented status of some specific particles.

7. Line 342-344: "*In addition, ice-ice collisions is believed to be most efficient at colder temperature (Takahashi et al., 1995) such as observed on this day. Therefore, we deduce that the ice-ice collisions were again the most likely active SIP mechanism in the low-level feeder cloud.*"
   This statement is based on the observations of ice particles that appear as fragments and association of these observations with ice-ice collision SIP mechanism. This induces the following questions: (a) Could the observed fragments be a result of anisoaxial ice particles impact with the HOLIMO walls? (b) Could the images of ice particles identified as fragments belong to intact particles? (c) Could the observed SIP particles originate from other SIP mechanisms?

**Other comments**

1. Line 203. *"…and drizzle drops (defined with diameter larger than 64 μm) were observed during four measurement flights"*. It is worth indicating maximum droplet size $D_{max}$ for each of those four flights.

2. Figure 7c. It is worth adjusting colors for a better reading of the diagram in Fig.7c. After printing, brown and violet appear to be too close, and they can be confused with each other.

3. Line 346 *"…and a volume of 6425 L."* It is worth indicating whether this number refers to the total or only in-cloud sampling.

Alexei Korolev

---

## Author Response (AR1)

**Review for: Conditions favorable for secondary ice production in Arctic mixed-phase clouds by Julie T. Pasquier**

**1   Summary**

**Pasquier et al. investigate the conditions that favor secondary ice production (SIP) in Arctic clouds, observed during NASCENT campaign, and examine the possible underlying mechanisms. For their investigations they use in-situ cloud microphysical measurements derived by a holographic cloud imager on a tethered balloon system and ground-based remote-sensing and ice nucleating particle (INP) measurements. Their analysis focuses on a six-day period which revealed the occurrence of SIP for about 40% of the time, while very high SIP events (with ICNCs > 10 L-1 ) were identified in 3.5% of the analyzed data. The highest SIP efficiency was found at temperatures between -3o C and -5o C; interestingly, this was attributed to the drop-shattering mechanism rather than to the Hallett-Mossop process, which has been thought to be the dominant SIP mechanism at these temperatures. Ice-ice collisions is also identified as the second most important SIP mechanism, especially at colder temperatures down to -24°C. This is a very well-written paper and a very interesting study. Models fail to reproduce the micro- and macro- physical structure of Arctic clouds and the description of ice microphysical processes has long been known to be a main contributor to these errors. Also while SIP has been hypothesized to be responsible for the enhanced cloud ice number concentrations often observed in the pristine Arctic environment, where INP availability is limited, the exact SIP mechanisms remain unknown. In general, there are very few in-cloud ICNC datasets from the Arctic and also most of them are not combined with INP measurements, which makes it difficult to quantify the influence of primary ice production versus SIP. This highlights the importance of the present study and the analyzed datasets for understanding ice production in Arctic clouds. For this reason, I recommend this paper for publication after the comments below have been addressed.**

*We would like to thank the anonymous reviewer for taking the time to carefully review our manuscript and provide insightful feedback that has improved the quality of our manuscript. Each of the comments are addressed point-by-point in bold font and all of the changes in the text and their associated line numbers are posted below.*

*We would like to point out that additional validation checks on the holographic cloud microphysical data were done to improve the data quality. Holograms with too many bright pixels, which may have inhibited the correct detection of cloud particles, were not considered in the data analysis (approximately 5 % of the holograms). In addition, the frequency distributions of the spatial position of the particle in the detection volume were analysed. The removal of the cluster of artifacts reduced the concentration of water droplets with D<40 μm. For this reason, the exact numbers in the revised manuscript may differ from the original manuscript and the Fig. 3 and Fig. 6-12 were updated, as well as Table 1 were updated, while the interpretation and conclusions remain unchanged by this clean-up.*

**2    Comments:**

**Line 63-64: maybe discuss a bit how ice shape is expected to influence SIP (with references)**

*Thank you for this comment. We have reformulated the sentences on lines 65-67 of the revised manuscript: "In particular, large rimed ice crystals were found to increase the rate of splinters ejected during rime-splintering (Hallett and Mossop, 1974) and ice-ice collision (Vardiman, 1978). Particles with complex shapes are more likely to produce fragments during sublimation (Bacon et al., 1998)."*

**Line 105: since measurements were collected during four months in total, why only such a small sample of six days is presented here? Please explain**

*During the first campaign in October-November 2019, more flights were performed with HoloBalloon. However, the data analysis of holographic data is computationally expensive and time consuming, thus we limited the analysis to measurements days that we consider the most interesting in this study. Additionally, the limited reachable flight altitude (1000 m) of the kytoon and the harsh Arctic conditions frequently experienced in spring 2020 unfortunately limited the collected sample during the second campaign in March-April. We hope that in the future more robust sampling systems for these conditions will be developed.*

**Line 111-114: are particles below 25 micro re-examined manually or are they treated as droplets in the analysis? In the case they are treated as droplets, can you estimate the magnitude of SIP underestimation? Fragments generated by drop-shattering can about 10 micro (Phillips et al., 2018)), while these can be even smaller for the other two SIP processes.The same question concerns misclassified ice crystals with circular shape.**

*Due to the 3 micron pixel size of HOLIMO, determining the shape of anything smaller than 25 microns is extremely difficult (Ramelli et al., 2020). Therefore, all particles smaller than 25 microns are classified as cloud droplets. Consequently some small ice crystals may be misclassified as cloud droplets. However, as these small ice crystals are expected to grow to at least 25 microns in relatively short time periods, e.g. within 100 seconds for columns at the warmest temperatures (Korolev et al., 2020), and as these particles have very slow fall velocities, the concentration of secondary ice particles quantified in this study is likely representative of the mean cloud state. At the same time, any circular ice crystals larger than 25 microns, are likely frozen droplets that have not become faceted yet and these would indeed be classified as cloud droplets. Although these droplets may have contributed to the production of secondary ice due to droplet freezing and shattering (e.g., Lauber et al., 2018) they are not technically secondary ice particles themselves. This means that any frozen droplets that have not become faceted and are still classified as cloud droplets should not be considered secondary ice and therefore, their classification as cloud droplets is not influencing the secondary ice concentrations. As mentioned in the text, any cloud particle classified as an ice crystal by the neural network (Touloupas et al. (2020)) and is larger than 25*

*microns is then hand-labeled and confirmed as ice or not.*

**Line 114-115: could you provide more details on the criteria (characteristics) used to classify to ice particles as recirculated or aged? How do you separate these two categories?**

*Thank you for raising this point. As described in Section 3.2, recirculation particles were growing in different preferential temperature growth regimes (plates or columnar). This gives them more than one primary habit, e.g. capped columns with columns growing out of the plate corners. In contrast, aged ice crystals are rimed, aggregates or have irregular shapes and thus their form was likely shaped by interacting with other particles. One manuscript was submitted to GRL describing in detail the formation of the recirculation particles (Pasquier et al., 2022b). To make the difference between aged and recirculation particles clearer, we have reformulated the sentence on line 115-118 of the revised manuscript to : "All ice crystals were manually classified into habits based on their 2D shape to plates, columns, frozen drops, recirculation particles showing evidences for growth in the plate and columnar growth regimes (see Section 3.2 and Pasquier et al. (2022b) for details), and aged particles that comprise rimed, aggregated, and irregular ice crystals."*

**Lines 125-126: what do you mean 'minimized'? Could you provide an estimate for how frequently shattering occurs? You should use the inter-arrival time algorithm (Korolev and Field, 2015) to identify shattering artifacts and exclude them from the analysis.**

*Thank you for this comment. We totally agree that using an inter-arrival time algorithm is a really nice technique for eliminating the shattered particles in 2D-optical array probes. However, unlike the quasi-continuously measurements of 2D-optical array probes, holographic instruments only capture an image at a fixed frame rate e.g. 6 Hz. From a single image, the properties of all the particles (up to 1000) inside the sample volume are retrieved. This means that there is no inter-arrival time between particles as all of the particles in the sample volume are captured simultaneously. Nevertheless, to reduce any potential impact of shattering, the reconstructed sample volume is selected such that the volume within 2 cm of the tips is excluded from the analysis. Additionally, we analyzed the frequency distribution of the particles in the 3D sample volume and could not identified clusters of particles in the sample volume, which could be associated with shattered particles, similar to particles with small inter-arrival times. Together with the fact that our measurements are taken at low wind speeds and without an inlet that particles can shatter on, we are convinced that shattering is minimized.*

**Lines 153-154: There is something I don't understand about this method. Why pristine ice crystals with size < 106 microns cannot be newly-formed primary ice crystals? Why these should solely be associated with SIP? Please explain.**

*Indeed, we fully agree that pristine ice crystals at these small sizes can also be formed via primary ice nucleation. However, when comparing the measured ice nucleating particle concentrations to the concentration of small ice crystals, we find that the small ice crystal concentrations are higher by orders of magnitude. Therefore, you are correct and some of the small of the pristine ice could be primary ice, but the fraction of these crystals is very small relative to those that must have formed*

*via secondary ice processes.*

**Lines 180-181: provide reference**

*Thank you for pointing this out. We have now added citations to Li et al. (2022) and Pasquier et al. (2022a) who present concurrent INP measurements that show an exponential relationship between INP concentration and temperature.*

**Lines 193-195: I am not convinced that the profiles derived on 8/11 and 11/11 are well- mixed. A $\theta$-gradient of 0.5 C is often used as criterion for decoupling (Sotiropoulou et al., 2014; Gierens et al., 2020). Please use one of the proposed methods in the literature to ensure cloud-surface coupling.**

*Thank you for this comment. We calculated the difference in potential temperature between the surface level and at the height halfway to the liquid base height, following the technique introduced by Gierens et al. (2020). The difference in potential temperature amounted to 0.65 K on 8.11.2019 and 0.63 K on 11.11.2019. If this difference is slightly above the 0.5 threshold used in the literature, we think that it is still correct to state that 'no strong decoupling case was observed'.*

**Lines 238-239: you do not provide any information on updraft velocity for November 10 to support this statement. I suggest to provide a time-height cross-section for this parameter (at least in the appendix or as supplementary material).**

*Thank you for pointing this out. We were not giving the updraft velocities here, as we think that the wind speed and direction information shown in Figs. 3b and A1 are sufficient to prove that the dynamics were weak. Indeed, we can see from these Figures that the wind speed was below 5 m s$^{-1}$ and that no wind shear was prevailing in the sampled cloud. We have however added information about the updraft speed on lines 237-238 of the revised manuscript: "The dynamic was weak within this cloud, as the horizontal and vertical wind speeds did not exceed 5 m s$^{-1}$ (Fig. 3b) and 1 m s$^{-1}$, respectively."*

**Section 3.2: While this section focuses on the investigation of the high SIP event in the afternoon of 11/11, it is worth including a short discussion for the possible drivers of the weaker SIP before 18:00.**

*Thank you for this comment. We agree that it is interesting to discuss potential SIP mechanism for the weaker SIP before 18:00 UTC. We have now added the information on lines 276-278: "Ice crystals observed before 18:00 UTC are largely aged particles, whereas ice crystals observed during SIP periods starting from 18:10 UTC were frozen drops, recirculated particles..." and on lines 279-281: "The presence of aged particles together with cloud droplets smaller than 12 μm and larger than 24 μm before 18:00 UTC suggests that the rime-splintering process could be responsible for the ICNC$_{pr<106\,μm}$ below 5 L$^{-1}$. "*

**Lines 332-333: However, peaks in the columnar ice concentrations before 13:00, which are of similar magnitude as the one observed round 13:00-13:15, are not associated with CDNCs**

**increases. What is the reason behind their formation?**

*Thank you for raising this point. The peaks in $ICNC_{pr<106\,\mu m}$ before 13:00 UTC are of similar magnitude as the one at 13:10-15 UTC, however columns and plates contribute to approximately the same extend to the increase, whereas at 13:10 and 14:00 UTC the columns clearly dominate the increase in $ICNC_{pr<106\,\mu m}$.*

**Line 333: There seems to be an almost constant white shading at sizes between 7-8 micro in both Figures 6 and 8. Is this some kind of artifact?**

*Thank you for this comment. Yes, this was an artifact caused by the choice of bin sizes of the cloud droplet size distribution. We have now corrected this artifact in Figures 6 and 8.*

**Line 342: It worths discussing here a bit more about the ice-ice collision process. Is it the same mechanism here as proposed by Geogakaki et al. (2022) for seeder-feeder events? They suggest that ice particles falling from the upper cloud collide with ice particles within the lower mixed-phase layer, resulting in mechanical break-up. And if this the case, why the process does not take place earlier, since the seeding-feeding system is observed from approximately 12:00 to 14:00?**

*Thank you for this comment. First, we want to make clear that SIP was occurring during the entire flight, e.g. $ICNC_{pr<106\,\mu m}$ amounted up to 7 $L^{-1}$ at around 12:20 UTC (see Figure 9b). Second, we can not say with complete certitude that it was the same mechanism as in Georgakaki et al. (2022) occurring, but it is very likely that the particles falling from the upper cloud were contributing to the formation of secondary ice crystal by ice-ice collision. Therefore, we have now added the following on lines 350-352 of the revised manuscript: "We propose that the large ice crystals sedimenting from the seeder cloud are rapidly growing at lower altitude in the ice supersaturated regions. They could create secondary ice particles by colliding with other ice crystals in the low-level feeder cloud."*

**Line 344: please provide the reference of Georgakaki et al. (2022) here, who reached the same conclusion about seeder-feeder cases.**

*Thank you for pointing out the agreement with this study. We have now added the sentence on lines 353-354 of the revised manuscript: "This hypothesis is in agreement with the recent study by Georgakaki et al. (2022) associating the occurrence of the ice–ice collision mechanism with the occurrence of precipitating seeder–feeder events."*

**Line 344: do you exclude the possible contribution of sublimation break-up? Deshmukh et al. (2022) suggest that as precipitation particles falling from the seeder cloud into a sub-saturated environment may experience sublimation break-up. Then, as the new fragments enter the saturated conditions of the feeder cloud, they can further grow through vapor deposition and enhance ICNCs (see their schematic in Fig. 14).**

*Thank you for this comment. We agree with the argumentation of Deshmukh et al. (2022) that precipitation particles falling from the seeder cloud into a sub-saturated environment may experience sublimation break-up and enhance the ICNC in a low-level feeder cloud. However, these sublimating*

*ice crystals must still be large enough to overcome the updraft and sediment in the lower cloud and these ice crystals likely do not exhibits pristine shapes. As in this study we use the concentration of small pristine ice crystals to evaluate local SIP, ice crystals having experienced sublimation break-up during sedimentation would not be detected as SIP particles. We want to focus on the SIP occurring close to the measurement location in this study, and thus for consistency opted to exclude a discussion on all of the possible SIP process that could have been occurring in other parts of the atmosphere.*

**Line 361: Could the correlation between ice-snow concentrations be due to the fact that SIP particles will eventually grow to snowflakes? This means that increases in snow concentration would follow the increases in the number smaller ice particle.**

*Thank you for raising this point. In general you are correct that the snowflake-SIP particles connection suffers from the chicken-egg problem: the SIP particles growing to larger sizes could be used in the correlation. To prevent this, we use only snow crystals larger than 327 μm, as we state on lines 366-369 of the revised manuscript: "The analysis of the influence of ice crystals on SIP is delicate because it is possible that the larger ice crystals are secondary ice crystals having grown to larger sizes than the threshold used (106 μm). To overcome this issue, we discuss only the connection between SIP and ice crystals larger than 327 μm, and refer to these as snow crystals." Concerning the time lag between SIP particles and snow crystal concentration increase, one important fact to keep in mind, is that the ice crystal only stay associated with their environment of origin for a certain time period. Therefore, larger snow crystals are likely to have experienced horizontal and/or vertical advection or turbulent diffusion and to not all originate from the same cloud region. Therefore it is inappropriate to make conclusion about peaks in snow crystal concentrations following peaks in SIP.*

**Line 367-368: Again, the contribution of sublimation break-up should be investigated here (see comment above).**

*Thank you, as answered above, with our approach we only investigate SIP occurring close to the measurements location and therefore not the SIP at higher altitude.*

**Line 385: How sensitive are the results to the choice of this CDNC threshold?**

*Thank you for this comment. We calculated the OEF using CDNC threshold between 3 and 10 cm$^{-3}$. The results presented in Table1 below show that the frequency of occurrence is sensitive to the threshold used, but the OEF are not strongly sensitive to the CDNC threshold used. Indeed, for $SIP_{all}$, $SIP_{low}$, $SIP_{mod}$ the OEF are lower than 1 for all thresholds and for $SIP_{high}$ the OEF are higher than 1 for all thresholds.*

Table 1: Frequency of occurrence and OEF of the hydrometeor types cloud droplets with concentrations CDNC from 3, 5 and 8 cm $^{-3}$ as threshold during all measurements ($N_{all}$), $SIP_{all}$, $SIP_{low}$, $SIP_{mod}$, and $SIP_{high}$.

| | | $N_{all}$ | $SIP_{no}$ | $SIP_{all}$ | $SIP_{low}$ | $SIP_{mod}$ | $SIP_{high}$ |
|---|---|---|---|---|---|---|---|
| CDNC threshold=3 cm $^{-3}$ | F (%) | 47.15 | 50.36 | 42.05 | 45.61 | 34.32 | 58.82 |
| | OEF | | | 0.83 | 0.91 | 0.68 | 1.17 |
| CDNC threshold=5 cm $^{-3}$ | F (%) | 33.32 | 35.88 | 29.26 | 31.71 | 22.79 | 45.88 |
| | OEF | | | 0.82 | 0.88 | 0.64 | 1.28 |
| CDNC threshold=10 cm $^{-3}$ | F (%) | 20.65 | 23.29 | 16.47 | 18.05 | 10.99 | 32.94 |
| | OEF | | | 0.71 | 0.77 | 0.47 | 1.41 |

**Lines 386-387: This comment concerns the small OEF for cloud droplets. The presence of cloud drops is expected to be important for SIP within the Hallet-Mossop temperature zone, but not outside of it. If you calculate the OEF only for the H-M zone, does the factor changes significantly?**

*Thank you for this comment, this is an interesting point. Yes, if we calculate the OEF only the Hallett-Mossop temperature range, the OEF for the cloud droplet increase by a factor of 2 to 3 (see Table 2 below). However, one need to keep in mind that by taking only the HM temperature range into account, one is mostly looking at the two cases on 11 and 12 November, when high SIP events were occurring, but most likely because of droplet shattering and not of rime-splintering, as discussed in the manuscript.*

Table 2: Frequency of occurrence and OEF of the hydrometeor types cloud droplets in at all temperature ranges and only in the H-M temperature range.

| | | $N_{all}$ | $SIP_{no}$ | $SIP_{all}$ | $SIP_{low}$ | $SIP_{mod}$ | $SIP_{high}$ |
|---|---|---|---|---|---|---|---|
| CDNC at all temperatures | F (%) | 33.32 | 35.88 | 29.26 | 31.71 | 22.79 | 45.88 |
| | OEF | | | 0.82 | 0.88 | 0.64 | 1.28 |
| CDNC at H-M temperature | F (%) | 31.16 | 25.27 | 35.54 | 39.28 | 28.33 | 42.6900 |
| | OEF | | | 1.41 | 1.55 | 1.12 | 1.69 |

**Lines 400: updraft speeds are hardly discussed in this manuscript. I suggest to add a figure for this variable and discuss this more thoroughly in relevance to SIP occurrence and drizzle formation.**

*Thank you for this comment. We were investigating the influence of updraft and wind shear on SIP formation, as it is evident that updrafts are undeniably important for the formation of drizzle drops and large aged ice crystals that were found to be related to SIP. For this investigation we used the updrafts derived from the Doppler spectra as discussed in the supplementary material, as well as the updraft from the wind lidar (see Figure 1 below). However, we found no direct connection between updraft or wind shear and SIP. E.g., there was no evidence for an increase in updraft or*

*wind shear in regions with SIP. For example, on 11 November 2019, the high SIP regions were located in regions with moderate updrafts (1 $^{-1}$ to 2 m s$^{-1}$) and above the wind shear layer. On 12 November 2019 however, the high SIP regions were located in low updraft regions (<1 m s$^{-1}$) below the wind shear layer (see Figure 1 below).*

[Figure]

Figure 1: (a) Updraft velocities from the maximal Doppler velocity derived from the cloud radar Doppler spectra. The white line at 1200 m and the attenuated signal above is caused by a change in measurement settings of the cloud radar at this altitude. (b) Updraft velocities measured by the wind lidar. Note the different colorbar limit in (a) and (b). (c) Horizontal wind speed and direction measured by the wind lidar. (d) Wind shear derived from the wind lidar horizontal wind.

*A likely reason for the lack of connection between SIP and updraft or turbulence, is the insufficient quality and resolution of the wind measurements. Both techniques (wind lidar and Doppler spectra) are not resolving updraft fluctuations or turbulence on the meter scale due to limitations in temporal and vertical resolution of the measurement. Additionally, the signal obtained from the wind lidar is attenuated by the presence of hydrometeors and wind speed measurements are lacking in the interior of the clouds or during strong precipitation events. The radar Doppler spectra -based technique relies on the slower edge of the Doppler spectrum to be representative of particles that*

*can act as tracers for air motion, and it is influenced by the hydrometeor population and amount of turbulence. For these reason, we have omitted the discussion of the impact of wind and turbulence on SIP.*

**Line 433: Please add the reference of Luke et al (2021) here. They were the first to show that high SIP events are associated with the presence of large drops in Arctic clouds. It is worth trying to relate your analysis to their findings, derived with remote-sensing methods.**

*Thank you for this comment. We fully agree with you and have added the statement: "This would be in agreement with laboratory studies showing that a large number of splinters (>10) can be produced from the freezing of a single drop (Lauber et al., 2018; Korolev and Leisner, 2020) as well as with recent remote sensing studies showing that high SIP events are associated with the presence of large drops in Arctic clouds (Luke et al., 2021). " on lines 440-443 of the revised manuscript*

**Line 450: Could you infer a minimum INPC that is necessary to initiate SIP from your measurements? Or at least an INPC level below SIP is never expected to occur.**

*Thank you for this comment. It would be great to give a minimum INPC that is necessary to initiate SIP. Unfortunately, it results from this study that a unique threshold cannot be given, but that this threshold depends on other parameters such as the atmospheric dynamics or the presence of supercooled drizzle drops. Indeed, as the ice crystal formation was limited by the INPC on 10 November 2019, the INPC was not lower than on the other days, but the dynamic was likely inhibiting the initiation of SIP.*

**Line 461: This is a statement that is not clearly supported by the analyzed data. The connection of updraft velocities, CCN and drizzle concentrations should be shown explicitly in the figures.**

*Thank you for this comment. It is true that we don't show in figures showing explicitly the connection between updraft velocities, CCN and drizzle. The reason is that such a discussion would require a study on its own and is beyond the scope of this paper, which focuses on SIP. However, past studies have related CCN and updrafts to the formation of drizzle drops and a publication using data from the NASCENT campaign is currently in preparation for publication on this topic (Motos et al., in prep). To make more clear that we suggest that the formation of drizzle drops is linked to the low CCN and sufficient updraft but without proving this in the paper, we changed the sentence on line 469-470 of the revised manuscript to: "We suggest that the presence of SLD itself is related to the strong updrafts and low CCN concentrations observed in the clean Arctic environment.".*

**Lines 465-466: if the contribution of sublimation break-up cannot be excluded with the existing data, maybe the possibility of having more mechanisms activated should be addressed here.**

*Thank you for this suggestion. As discussed in the comment above, we only discuss the prevalence of SIP close of the measurement location and therefore we don't discuss the prevalence of sublimation breakup in particles precipitating from the seeder cloud.*

**References**

Bacon, N. J., Swanson, B. D., Baker, M. B., and Davis, E. J.: Breakup of levitated frost particles, Journal of Geophysical Research: Atmospheres, 103, 13 763–13 775, https://doi.org/https://doi.org/10.1029/98JD01162, 1998.

Deshmukh, A., Phillips, V. T. J., Bansemer, A., Patade, S., and Waman, D.: New Empirical Formulation for the Sublimational Breakup of Graupel and Dendritic Snow, Journal of the Atmospheric Sciences, 79, 317 – 336, https://doi.org/10.1175/JAS-D-20-0275.1, 2022.

Georgakaki, P., Sotiropoulou, G., Vignon, E., Billault-Roux, A.-C., Berne, A., and Nenes, A.: Secondary ice production processes in wintertime alpine mixed-phase clouds, Atmospheric Chemistry and Physics, 22, 1965–1988, https://doi.org/10.5194/acp-22-1965-2022, 2022.

Gierens, R., Kneifel, S., Shupe, M. D., Ebell, K., Maturilli, M., and Löhnert, U.: Low-level mixed-phase clouds in a complex Arctic environment, Atmospheric Chemistry and Physics, 20, 3459–3481, https://doi.org/10.5194/acp-20-3459-2020, 2020.

Hallett, J. and Mossop, S. C.: Production of secondary ice particles during the riming process, Nature, 249, 26–28, https://doi.org/10.1038/249026a0, 1974.

Korolev, A. and Field, P. R.: Assessment of the performance of the inter-arrival time algorithm to identify ice shattering artifacts in cloud particle probe measurements, Atmospheric Measurement Techniques, 8, 761–777, https://doi.org/10.5194/amt-8-761-2015, 2015.

Korolev, A. and Leisner, T.: Review of experimental studies on secondary ice production, Atmospheric Chemistry and Physics Discussions, 2020, 1–42, https://doi.org/10.5194/acp-2020-537, 2020.

Korolev, A., Heckman, I., Wolde, M., Ackerman, A. S., Fridlind, A. M., Ladino, L. A., Lawson, R. P., Milbrandt, J., and Williams, E.: A new look at the environmental conditions favorable to secondary ice production, Atmospheric Chemistry and Physics, 20, 1391–1429, https://doi.org/10.5194/acp-20-1391-2020, 2020.

Lauber, A., Kiselev, A., Pander, T., Handmann, P., and Leisner, T.: Secondary Ice Formation during Freezing of Levitated Droplets, Journal of the Atmospheric Sciences, 75, 2815–2826, https://doi.org/10.1175/JAS-D-18-0052.1, 2018.

Li, G., Wieder, J., Pasquier, J. T., Henneberger, J., and Kanji, Z. A.: Predicting atmospheric background number concentration of ice nucleating particles in the Arctic, Atmospheric Chemistry and Physics Discussions, pp. 1–29, https://doi.org/https://doi.org/10.5194/acp-2022-21, 2022.

Luke, E. P., Yang, F., Kollias, P., Vogelmann, A. M., and Maahn, M.: New insights into ice multiplication using remote-sensing observations of slightly supercooled mixed-phase clouds in the Arctic, Proceedings of the National Academy of Sciences, 118, https://doi.org/10.1073/pnas.2021387118, 2021.

Motos, G., Georgakaki, P., Wieder, J., Aas, W., Lunder, C., Freitas, G., Krejci, R., Mohr, C., Zieger, P., Lohmann, U., and Nenes, A.: Aerosol size distribution, hygroscopicity and cloud formation from fall to spring at an Arctic mountain site, in prep.

Pasquier, J. T., David, R. O., Freitas, G., Gierens, R., Gramlich, Y., Haslett, S., Li, G., Schäfer, B., Siegel, K., Wieder, J., Adachi, K., Belosi, F., Carlsen, T., Decesari, S., Ebell, K., Gilardoni, S., Gysel-Beer, M., Henneberger, J., Inoue, J., Kanji, Z. A., Koike, M., Kondo, Y., Krejci, R., Lohmann, U., Maturilli, M., Mazzolla, M., Modini, R., Mohr, C., Motos, G., Nenes, A., Nicosia, A., Ohata, S., Paglione, M., Park, S., Pileci, R. E., Ramelli, F., Rinaldi, M., Ritter, C., Sato, K., Storelvmo, T., Tobo, Y., Traversi, R., Viola, A., and Zieger, P.: The Ny-Ålesund Aerosol Cloud Experiment (NASCENT): Overview and First Results, Bulletin of the American Meteorological Society, https://doi.org/10.1175/BAMS-D-21-0034.1, 2022a.

Pasquier, J. T., Henneberger, J., Ramelli, F., Korolev, A., Wieder, J., Lauber, A., Li, G., David, R. O., Carlsen, T., Gierens, R., Maturilli, M., and Lohmann, U.: Understanding the history of complex ice crystal habits deduced from a holographic imager, Geophysical Research Letters, submitted, 2022b.

Phillips, V. T. J., Patade, S., Gutierrez, J., and Bansemer, A.: Secondary Ice Production by Fragmentation of Freezing Drops: Formulation and Theory, Journal of the Atmospheric Sciences, 75, 3031 – 3070, https://doi.org/10.1175/JAS-D-17-0190.1, 2018.

Ramelli, F., Beck, A., Henneberger, J., and Lohmann, U.: Using a holographic imager on a tethered balloon system for microphysical observations of boundary layer clouds, Atmospheric Measurement Techniques, 13, 925–939, https://doi.org/10.5194/amt-13-925-2020, 2020.

Sotiropoulou, G., Sedlar, J., Tjernström, M., Shupe, M. D., Brooks, I. M., and Persson, P. O. G.: The thermodynamic structure of summer Arctic stratocumulus and the dynamic coupling to the surface, Atmospheric Chemistry and Physics, 14, 12 573–12 592, https://doi.org/10.5194/acp-14-12573-2014, 2014.

Touloupas, G., Lauber, A., Henneberger, J., Beck, A., and Lucchi, A.: A convolutional neural network for classifying cloud particles recorded by imaging probes, Atmospheric Measurement Techniques, 13, 2219–2239, https://doi.org/10.5194/amt-13-2219-2020, 2020.

Vardiman, L.: The Generation of Secondary Ice Particles in Clouds by Crystal–Crystal Collision, Journal of Atmospheric Sciences, 35, 2168 – 2180, https://doi.org/10.1175/1520-0469(1978)035<2168:TGOSIP>2.0.CO;2, 1978.

**Review of "Conditions favorable for secondary ice production in Arctic mixed-phase clouds" by Pasquier et al.**

**1  Overview**

The paper presents an important in-situ observation of secondary ice production in Arctic clouds from a tethered balloon with the help of a high-resolution imaging probe HOLIMO. The HOLIMO provided photographic quality images of cloud particles, which, in the majority of cases, allowed for unambiguous identification of particles habit and their phase. The in-situ observations were complemented by remote sensing observations, which allowed for accurate positioning of the HOLIMO measurements with respect to cloud boundaries. Ground based measurements of INPs provided a significant contribution to the value and quality of the collected data set. One of the major outcomes of this study is lowering the threshold concentration of ice particles separating primary and secondary ice production. This is different from the past observations when SIP was identified as an explosive increase in the number concentration of ice particles exceeding hundreds and thousands per liter. In the present study, the threshold concentration was reduced to the order of 0.1-10 per liter. The paper provides an important contribution to the understanding of the role of SIP in ice formation and, undoubtedly, it should be published in ACP. The paper is well written, and I do not have any significant comments on this work. There are a few minor comments which are worth addressing prior to publication.

**Recommendation: The paper should be published in ACP after minor revisions.**

*We would like to thank Dr. Korolev for his encouraging remarks and insightful comments for improving the manuscript. We have addressed his comments point-by-point below in italic font and have included the changes in the text and their line numbers in the revised manuscript.*
*We would like to point out that additional validation checks on the holographic cloud microphysical data were done to improve the data quality. Holograms with too many bright pixels, which may have inhibited the correct detection of cloud particles, were not considered in the data analysis (approximately 5 % of the holograms). In addition, the frequency distributions of the spatial position of the particle in the detection volume were analyzed. The removal of the cluster of artifacts reduced the concentration of water droplets with D<40 μm. For this reason, the exact numbers in the revised manuscript may differ from the original manuscript and the Fig. 3 and Figs. 6-12, as well as Table 1 were updated, while the interpretation and conclusions remain unchanged by this clean-up.*

**2  Comments**

1. **Line 204 and throughout the text: "drizzle drops (defined with diameter larger than 64 μm)" Following commonly accepted definition (e.g., Glossary of Meteorology), drizzle is defined as drops in the size range 100 μm<D<500μm. The authors may consider using the term "supercooled large droplets" (SLD) instead. SLD is defined as droplets with D>50 μmm**

**at T<0C.**

*Thank you for this comment. We have replaced the term drizzle drops with supercooled large droplets (SLD) throughout the manuscripts, but amended its definition according to our measurement size bins as droplets > 64 μm at temperature below 0 °C.*

**2. Line 124-126: "In addition, thanks to the low true air speed of HOLIMO on the tethered balloon system and the adequate tower tips, the shattering of ice crystals in the sample volume is minimized." Reduction of the sampling speed will undoubtedly reduce the effect of shattering of ice due to decreasing the kinetic energy of particles impact. However, the HOLIMO inlet axis is unlikely to be always perfectly aligned with the local airflow, and particles with anisoaxial trajectories are likely to be frequently present during sampling. This is specifically relevant to the cases with the turbulent environment. Particle fall speed will also contribute to the deviation of the particle trajectories from the inlet flow. Such particles will impact with the HOLIMO inlet walls at a speed of ~1-2m/s, and they may get fragmented and contaminate measurements shattering artifacts. Fragmentation of freefalling (1-2m/s) particles on impact with a solid surface of observed by Vardiman (JAS, 1978). In this regard, it would be reasonable to make a disclaimer when talking about the effect of shattering.**

*Thank you for raising this point. The version of HOLIMO used on the HoloBalloon platform does not have an inlet (see Ramelli et al. (2020) and Pasquier et al. (2022)) and the imaging arms are oriented horizontally (see Figure 1 below). Therefore, the shattering of falling ice crystals on the inlet can be excluded from this study. Moreover, we analyzed the frequency distribution of the particles in the 3D sample volume and could not identify clusters of particles in the sample volume, which could be associated with shattered particles, similar to particles with small inter-arrival times. Finally, the kytoon and the payload are automatically orienting in the wind direction with the wind vane. To make this clearer in the text we have now added the following on lines 127-129: "In addition, HOLIMO's open-path configuration and antishattering tips (as recommended in Korolev et al. (2013)) as well as the automatic orientation of the kytoon and the payload in the wind direction mitigate the shattering of falling ice crystals in the sample volume."*

[Figure]

Figure 1: Payload hanging below HoloBalloon with description of the different instrumental parts.

**3. Line 158-159: "Non-pristine crystals cannot have formed from vapor deposition growth, and could originate from breakups during impact with the instrument payload or from rime falling from the tethered balloon." The first part of this statement should be corrected. For example, non-pristine ice particles can be formed during water vapor deposition growth on polycrystalline frozen droplets or developed after formation dislocation in the ice crystal lattice of pristine particles.**

*We fully agree with you. As such we have now rephrased the sentence to state: "We exclude non-pristine ice crystals from the SIP analysis as their habits and thus, the environment in which they grew in cannot unambiguously be defined. This also removes the potential for any falling ice or rime from the balloon to be misclassified as SIP." on line 161-163 of the revised manuscript.*

**4. Line 290: "Indeed, particles resembling broken branches were observed (highlighted with the dark brown box in Figure 7a)." Could you elaborate why the images of the boxed particles were identified as broken branches. Particle having such shape could be formed without fragmentation.**

*Thank you for pointing this out. We fully agree and rather meant to give examples of the fragile looking nature of the needles growing on the recirculation particles that could break off and lead to additional SIP. As this is already stated in the previous sentence, we have now removed this sentence from the text and the brown box in Figure 7a.*

**5. Line 320: "The CDNCs measured by HOLIMO was generally below 1 cm$^{-3}$ except at 13:10 UTC or between 13:45 and 14:15 UTC, when increases in CDNC were observed (Fig. 8b). These comparatively large CDNCs (> 10 cm$^{-3}$) are observed when HoloBalloon was in**

the transition region from low to high radar reflectivity (i.e. in the embedded supercooled liquid layer). It suggests that in this region water saturation was sustained and promoted the formation and growth of cloud droplets, while below, the environment was subsaturated with respect to water and the cloud droplets were evaporating." I have serious concerns regarding the existence of sustainable liquid clouds in the boundary layer with droplets D<40μm and number concentration <1cm$^{-3}$ (e.g., as shown in Fig.8b). Such clouds have a high level of instability and may exist in a relatively turbulent environment only for a limited time since any vertical motion will result in activation of CCNs or complete evaporation of droplets. Therefore, the interpretation of observations of cloud segments with N<0.1cm$^{-3}$ as liquid or mixed-phase causes a question of whether the phase of particles with D<40 μm was identified correctly, or the images of these particles are a result of some non-recognized artifacts. In relation to Fig 8, I would like to add that the rapid increase of the radar reflectivity from -10dBZ to ~10dBZ at approximately 1km altitude is indicative of a presence of a liquid layer there. Such layers usually result in the enhanced growth of seeded from above ice crystals and an increase in radar reflectivity. On the other hand, Arctic clouds are frequently decoupled. I am wondering if the cloud base identified from the ceilometer measurements, as shown in Fig.8a, corresponds to the lower layer. This may result in a perception that the HOLIMO measurements were performed inside a liquid/mixed-phase cloud layer, whereas the measurements in fact were sampled in between cloud layers. Could you also comment on this?

*Thanks for raising this point. With respect to the comment about artifacts potentially influencing the identification of the droplets D<40 μm. We would like to point out that after cleaning-up the dataset as mentioned above, the CDNC is reduced between the peaks at 13:10 and between 13:45 and 14:15 UTC. We agree with the reviewer that the regions with CDNC < 1cm$^{-3}$ cannot be considered as mixed-phase or liquid clouds. While validating the dataset, we ensured that the detected cloud droplets are not generally wrongly artifacts classified as cloud droplets. However, as we do not differentiate between cloud droplets and ice crystals below 25 μm, smaller (sublimating) ice crystals are also classified as cloud droplets. Therefore we cannot tell if the detected particles below 25 μm are cloud droplets or ice crystals. We thus agree that in the regions where these low CDNCs are observed, HoloBalloon is likely measuring below the liquid layer. Meanwhile at 13:10 UTC or between 13:45 and 14:15 UTC when the higher CDNCs are observed, HoloBalloon is measuring inside of the liquid layer. This is consistent with the ceilometer observations in Fig. 8a where the track of HoloBalloon is above the detected cloud base. Unfortunately, the cloud base detection is obscured during heavier precipitation and therefore, it is difficult to always determine where the lowest liquid cloud base is. This makes identifying the presence of multiple liquid layers especially challenging. However, as the high CDNCs were only observed when HoloBalloon was at its highest altitudes, it is unlikely that several liquid layers were present between the ground and maximum height of HoloBalloon. Regardless, we intended for this section to state that HoloBalloon is measuring in the mixed phase/liquid layer during the high CDNC periods (e.g. CDNC>10 cm$^{-3}$) and that it is below this layer when the lower concentrations are observed. We have now reformulated this section to make this clearer on lines 325-329 by stating: "The CDNCs measured*

*by HOLIMO only reached concentrations higher than 10 cm⁻³ at 13:10 UTC and between 13:45 and 14:15 UTC (Fig. 8b). These higher CDNCs (> 10 cm⁻³) are observed when HoloBalloon was in the transition region from low to high radar reflectivity (i.e. in the embedded supercooled liquid layer). It suggests that in this region water saturation was sustained and promoted the formation and growth of cloud droplets, while below, in the regions with low CDNCs, the environment was subsaturated with respect to water and the cloud droplets were evaporating."*

**6. Line 341-342: "On the contrary, some ice crystals showed broken features, as highlighted by the blue frames in Figure 9a. As the ICNCs were large (up to 55 L⁻¹) collisions between ice crystals have likely occurred." I would be conservative regarding the interpretation of the boxed images in Fig.9a as fragments. Images of ice particles with under- or non-developed corners or branches can be found in Nakaya (1954) or Bentley and Humphreys (1931), respectively. Could you comment or provide strong evidence supporting the fragmented status of some specific particles.**

*Thank you for pointing this out. First, we have now updated the text to read that the example images are framed in black boxes, not blue, as is shown in the Figure. Second, you are correct, we cannot definitively state that the underdeveloped corners pointed to, are due to fragmenting. To account for this, we have now weakened the statement on lines 346-348 of the revised manuscript to read: "On the contrary, some ice crystals contained underdeveloped corners (highlighted by the black frames in Figure 9a.), which could be a result of recent ice-ice collisions. As the ICNCs were large (up to 78 L⁻¹) collisions between ice crystals likely occurred."*

**7. Line 342-344: "In addition, ice-ice collisions is believed to be most efficient at colder temperature (Takahashi et al., 1995) such as observed on this day. Therefore, we deduce that the ice-ice collisions were again the most likely active SIP mechanism in the low-level feeder cloud." This statement is based on the observations of ice particles that appear as fragments and association of these observations with ice-ice collision SIP mechanism. This induces the following questions: (a) Could the observed fragments be a result of anisoaxial ice particles impact with the HOLIMO walls? (b) Could the images of ice particles identified as fragments belong to intact particles? (c) Could the observed SIP particles originate from other SIP mechanisms?**

*Thank you for these pertinent questions. (a) As discussed in comment 2, we do not believe that shattering on the probe arms is a likely possibility as HOLIMO has no inlet and antishattering tips. Moreover, we examined the data for potential contamination from rime falling from the rope or from the kytoon in the measurement volume and removed the possibly contaminated holograms. (b) As discussed in the previous comment 6, we cannot be sure that the underdeveloped corners in the example images shown in Fig. 9a are due to collisions and therefore rephrased these sentence as mentioned in the answer of comment 6. However, the observation of small pristine ice crystals with concentrations up to 19 L⁻¹ indicates that SIP was occurring. (c) Droplet shattering, and rime splintering can be ruled out due to the lack of SLD and temperatures far below the HM temperature range. We would like to refer here to lines 343-346 of the revised manuscript: "...the observed*

*temperature (-24 ° to -18°C) was far below the temperature range of rime splintering (-8°C to -3°C). Furthermore, no large droplets necessary for the droplet shattering process were observed. Therefore, the rime-splintering and the droplet shattering processes are unlikely to have played a significant role as SIP mechanisms in the observed cloud." One other possibility may be the occurrence of the thermal shock process (Korolev and Leisner, 2020), but because of the low CDNC this process which requires the collision between cloud droplets and ice crystals is unlikely to play a significant role in this cloud. Therefore, we believe that ice-ice collision is the most likely active SIP process. We have elaborated on this hypothesis on lines 350-354 of the revised manuscript: "We propose that the large ice crystals sedimenting from the seeder cloud are rapidly growing at lower altitude in the ice supersaturated regions. They could create secondary ice particles by colliding with other ice crystals in the low-level feeder cloud. This hypothesis is in agreement with the recent study by Georgakaki et al. (2022) associating the occurrence of the ice–ice collision mechanism with the occurrence of precipitating seeder–feeder events."*

**References**

Georgakaki, P., Sotiropoulou, G., Vignon, E., Billault-Roux, A.-C., Berne, A., and Nenes, A.: Secondary ice production processes in wintertime alpine mixed-phase clouds, Atmospheric Chemistry and Physics, 22, 1965–1988, https://doi.org/10.5194/acp-22-1965-2022, 2022.

Korolev, A. and Leisner, T.: Review of experimental studies on secondary ice production, Atmospheric Chemistry and Physics Discussions, 2020, 1–42, https://doi.org/10.5194/acp-2020-537, 2020.

Korolev, A., Emery, E., and Creelman, K.: Modification and Tests of Particle Probe Tips to Mitigate Effects of Ice Shattering, Journal of Atmospheric and Oceanic Technology, 30, 690 – 708, https://doi.org/10.1175/JTECH-D-12-00142.1, 2013.

Pasquier, J. T., David, R. O., Freitas, G., Gierens, R., Gramlich, Y., Haslett, S., Li, G., Schäfer, B., Siegel, K., Wieder, J., Adachi, K., Belosi, F., Carlsen, T., Decesari, S., Ebell, K., Gilardoni, S., Gysel-Beer, M., Henneberger, J., Inoue, J., Kanji, Z. A., Koike, M., Kondo, Y., Krejci, R., Lohmann, U., Maturilli, M., Mazzolla, M., Modini, R., Mohr, C., Motos, G., Nenes, A., Nicosia, A., Ohata, S., Paglione, M., Park, S., Pileci, R. E., Ramelli, F., Rinaldi, M., Ritter, C., Sato, K., Storelvmo, T., Tobo, Y., Traversi, R., Viola, A., and Zieger, P.: The Ny-Ålesund Aerosol Cloud Experiment (NASCENT): Overview and First Results, Bulletin of the American Meteorological Society, https://doi.org/10.1175/BAMS-D-21-0034.1, 2022.

Ramelli, F., Beck, A., Henneberger, J., and Lohmann, U.: Using a holographic imager on a tethered balloon system for microphysical observations of boundary layer clouds, Atmospheric Measurement Techniques, 13, 925–939, https://doi.org/10.5194/amt-13-925-2020, 2020.

---

## Author Response (AR2)

**Editor decision: Publish subject to minor revisions (review by editor)**

**Comments to the author:**

**Dear Dr. Pasquier,**

**I would like to thank you for submission of the revised manuscript, and also the two reviewers for their careful read and comments that led to substantial improvements. The use of HOLIMO to relate the freezing of supercooled large droplet to high concentrations of small ice particles is an important contribution to the field. That SLDs and high ice crystal concentrations appear to go hand-in-hand has long been recognized but the explanation has been somewhat mysterious. In my view this paper will help other researchers solidify their understanding of the problem.**

**To this end, there are two things that I would like to see included in the document, although I leave it up to the authors as to whether either is done. The first is that it appears that INPs are not necessarily a primary mechanism determining ice crystal concentrations in the Arctic. It suggests the focus of future study should perhaps be secondary ice crystal production mechanisms. More emphatic statements to this effect in the abstract and conclusions may be warranted.**

**Second item that may be desirable is to do some back-of-the-envelope calculations to determine what magnitude and rate of splinter production would be required to account for the HOLIMO observations. For example, line 285 to 289 establish a "likely explanation" that looks like it could be easily tested or constrained with a simple differential equation leading to exponential growth subject to depletion of available SLDs. Supplementing a little mathematics with a cartoon might help readers better understand the jump from correlation to causation, such that parameterizations could eventually be developed.**

**I look forward to the revision, and also hope acknowledgment can be given in the paper to the reviewers.**

**Best regards,**

**Tim Garrett**

*Answer to the Editor Timothy Garrett*

*We would like to thank the editor Timothy Garrett for his encouraging remarks and helpful comments, which helped improving the manuscript. We have addressed his two comments below and have included the changes in the text and their line numbers in the revised manuscript. We want to apologize for having forgotten to acknowledge the reviewers, we included them in the acknowledgment section of the revised manuscript.*

*Regarding the first point, we agree that it is worth mentioning that even if INPs are necessary for the formation of the first primary ice crystals, SIP is controlling the ICNC when it is active. As such it should be prioritized in future studies which aim to quantify the concentration and evolution of ICNC in Arctic MPCs. We have therefore increased the emphasis around this point in the abstract on lines 19-20: "Despite the undeniable necessity of INPs for the formation of the first ice crystals, the extent to which SIP occurs when activated determines the ice crystal number concentration.", as well as in the conclusion on lines 482-485: "Although INPs are necessary for the formation of the first (primary) ice crystals, our results indicate that, when SIP processes are active, they ultimately determine the ICNC. Therefore, the focus of future work investigating the evolution of ice crystal concentrations in Arctic low-level clouds should be placed on SIP."*

*Regarding the second point, we did some calculations concerning the high SIP case on 11 November 2019 and added this information to strengthen our hypothesis on lines 293-295: "With SLDNC of about 50 $L^{-1}$ and a frozen drop concentration reaching up to 6 $L^{-1}$, around 10% of the SLDs seem to have frozen, thereby producing on average approximately 15 secondary ice crystals.".*
*However, it is unfortunately not possible to infer causation with certitude for a specific SIP mechanism occurring in natural clouds, as one does not observe or measure SIP occurring directly, but only its subsequent result: high concentration of small pristine ice crystals. In the cases observed in this study, it is also difficult to exclude the simultaneous occurrence of several SIP processes. Therefore, one would first need to identify the cases where SIP is occurring and then differentiate between the SIP mechanism active in order to come up with a calculation of the number of ice crystals produced from each SIP mechanism. For example, in the above mentioned case where droplet shattering is a 'likely explanation', one can not completely exclude the occurrence of the Hallett-Mossop process. Only taking the SLDNC into account in a parametrization would be inappropriate, as similar SLDNC were measured earlier during this measurement flight with much lower concentration of the small pristine ice crystals (1-3 $L^{-1}$) and almost no frozen droplets. Moreover, the droplet shattering mechanism is a cascading process (e.g., Lawson et al., 2015), which complicates the derivation of a parametrization.*
*For these reasons, the calculations are unfortunately not as easily derivable as suggested. We agree that it would be strongly beneficial to use this dataset to develop and/or test SIP parametrizations, but we think that deriving such a parametrization is beyond the scope of this manuscript. Nevertheless, we strongly encourage the scientific community to derive and test SIP parametrizations using this dataset which will be made publicly available.*

**References**

Lawson, R. P., Woods, S., and Morrison, H.: The Microphysics of Ice and Precipitation Development in Tropical Cumulus Clouds, Journal of the Atmospheric Sciences, 72, 2429 – 2445, https://doi.org/10.1175/JAS-D-14-0274.1, 2015.

---

## Author Response (AR3)

**Editor decision: Publish subject to technical corrections**

**Comments to the author:**

**Dear Dr. Pasquier,**
**I'm sorry, just one last technical correction. The English is cumbersome in the last sentence of the abstract, a sentence that I think is important and deserves to be written in the active voice. Also, INP isn't defined. My suggestion.**

**Although ice nucleating aerosol particles may be necessary for the initial freezing of water droplets, the ice crystal number concentration is frequently determined by secondary production mechanisms.**

**Regards,**

**Tim Garrett**

*Answer to the Editor Timothy Garrett*

*Dear Dr. Garrett,*

*We would like to thank you for this helpful comment. We agree with his comment and changed the last sentence of the abstract on lines 19-20 of the revised manuscript to: "Although ice nucleating aerosol particles may be necessary for the initial freezing of water droplets, the ice crystal number concentration is frequently determined by secondary production mechanisms."*

*Kind regards*
*Julie Pasquier*